# Switching action modes of miR408-5p mediates auxin signaling in rice

Fuxi Rong[1,2,6], Yusong Lv [1,2,6], Pingchuan Deng[1,3,6], Xia Wu[1], Yaqi Zhang[2], Erkui Yue[1], Yuxin Shen[1], Sajid Muhammad[1], Fangrui Ni[1], Hongwu Bian[4], Xiangjin Wei [5], Weijun Zhou[1], Peisong Hu[5] & Liang Wu [1,2] ✉

MicroRNAs (miRNAs) play fundamental roles in many developmental and physiological processes in eukaryotes. MiRNAs in plants generally regulate their targets via either mRNA cleavage or translation repression; however, which approach plays a major role and whether these two function modes can shift remains elusive. Here, we identify a miRNA, miR408-5p that regulates *AUXIN/INDOLE ACETIC ACID 30* (*IAA30*), a critical repressor in the auxin pathway via switching action modes in rice. We find that miR408-5p usually inhibits IAA30 protein translation, but in a high auxin environment, it promotes the decay of *IAA30* mRNA when it is overproduced. We further demonstrate that IDEAL PLANT ARCHITECTURE1 (IPA1), an SPL transcription factor regulated by miR156, mediates leaf inclination through association with miR408-5p precursor promoter. We finally show that the miR156-IPA1-miR408-5p-IAA30 module could be controlled by miR393, which silences auxin receptors. Together, our results define an alternative auxin transduction signaling pathway in rice that involves the switching of function modes by miR408-5p, which contributes to a better understanding of the action machinery as well as the cooperative network of miRNAs in plants.

MicroRNAs (miRNAs) are 20–24 nucleotide (nt) small RNAs that play fundamental roles in the regulation of gene expressions in eukaryotes. The primary miRNA unit in both animal and plants is typically transcribed by RNA polymerase II and forms a stem-loop transcript[1]. This hairpin precursor (pre-miRNA) is subsequently excised as an RNA duplex by an RNase III enzyme, Dicer or Dicer-like (DCL) protein, and generates two strands, one strand designated as miRNA*, which is degraded, while the other strand is called miRNA that enters ARGO-NAUTE (AGO)-mediated RNA-induced silencing complexes, where it scans complementary sequences and execute its functions[2].

Plant miRNAs have been thought to predominantly slice transcripts due to their strict base-pairing with targets in contrast to animal

miRNAs, which appear to be imperfectly matched with and mediate targets through translation inhibition. A couple of studies have shown that certain targets of miRNAs exhibit discrepancies between their transcripts and protein products, indicating that plant miRNAs have the capability to suppress target translation in addition to transcript degradation[3–7]. Furthermore, several factors that are specific to miRNA-entailed translational repression have been genetically characterized[3,8–12], supporting the notion that two action modes of miRNAs can be separated in plants. Nevertheless, because mRNA cleavage and protein translation inhibition seem to have similar sequence matching requirements between miRNA and targets in plants, and the decrease of mRNA gives rise to a reduction of protein, it

[1]National Key Laboratory of Rice Biology and Zhejiang Provincial Key Laboratory of Crop Germplasm Resources, College of Agriculture and Biotechnology, Zhejiang University, Hangzhou, Zhejiang 310058, China. [2]Hainan Yazhou Bay Seed Laboratory, Hainan Institute, Zhejiang University, Sanya, Hainan 572000, China. [3]State Key Laboratory of Crop Stress Biology in Arid Areas, College of Agronomy, Northwest A&F University, Yangling, Shaanxi 712100, China. [4]Institute of Genetics and Regenerative Biology, Key Laboratory for Cell and Gene Engineering of Zhejiang Province, College of Life Sciences, Zhejiang University, Hangzhou 310058, China. [5]National Key Laboratory of Rice Biology, China National Center for Rice Improvement, China National Rice Research Institute, Hangzhou, Zhejiang 310006, China. [6]These authors contributed equally: Fuxi Rong, Yusong Lv, Pingchuan Deng. ✉e-mail: liangwu@zju.edu.cn

is difficult to determine which regulatory mode of miRNA is more important, and whether miRNA can swap the two modes of target regulation is largely elusive[13,14].

Phytohormones play pivotal roles in almost all aspects of plant growth. As a key mediator of plant development, auxin is perceived by the receptors TRANSPORT INHIBITOR RESPONSE1 (TIR1)/AUXIN SIGNALING F BOX PROTEINS (AFBs), which leads to the activation of AUXIN RESPONSE FACTORS (ARFs) transcriptional factors through proteasomal degradation of a set of AUX/ INDOLE ACETIC ACID (IAA) family proteins[15]. Several miRNAs have been revealed to play roles in regulation of major steps in auxin signaling. For example, miR393, a conserved miRNA in diverse plants, cleaves *TIR1/AFB* transcripts and triggers the biogenesis of phased secondary siRNAs (phasiRNAs) to keep the homeostasis of Auxin cascade[16–18]. Thus, the networks involving miRNAs, phasiRNAs, mRNAs and proteins contribute to the robustness and flexibility to growth and development influenced by plant hormones.

One of the greatest challenges in crop improvement is that the trade-off between introducing a desirable trait in a new variety and potentially missing other beneficial characteristics[19]. However, miR156, a conserved small RNA molecule found in multiple plants, has emerged as an ideal for manipulating various important agronomic features through regulating a class of SQUAMOSA PROMOTER BINDING PROTEIN-LIKE (SPL) transcription factors[20–24]. Similarly, miR408 is also recently implicated as a promising candidate for crop improvement, since overexpression of miRNA408 has been shown to increase growth, photosynthesis and grain yield by down-regulating

*UCLACYANIN 8* (*UCL8*), a phytocyanin family gene, thereby impairing copper homeostasis in rice[25–27].

In previous study, we demonstrated that miRNA precursors in rice have the ability to generate multiple mature miRNAs simultaneously through the cooperative actions of different DCLs[28]. Similarly, *MIR408*, which is annotated in miRbase, is capable to produce two mature miRNAs, namely miR408-5p and miR408-3p (previously designated as miR408) in rice. Although the role of miR408-3p in growth and tolerance to stresses is clear, the targets and effects of miR408-5p on rice performance remain unknown. Here, we illustrated that miR408-5p is involved in auxin signaling in rice. Most strikingly, our finding reveals that miR408-5p usually inhibits target protein translation, but switches to mediate target mRNA cleavage when it is over-accumulated. Furthermore, we found that miR393 and miR156, two conserved miRNAs in plants, could hierarchically act with miR408-5p in the rice auxin pathway to control leaf inclination. Together, our results define a novel auxin transduction signaling pathway that is mediated by switching of function modes of miR408-5p.

## Results

### miR408-5p is abundant in rice

The rice genome contains a single *MIR408* locus, and the pre-miR408 has been annotated to generate two mature miRNAs, miR408-5p and miR408-3p, from each arm of the precursor (Fig. 1A). It is clear that miR408-3p is a conserved miRNA among different plant species and is therefore considered to be more abundant than miR408-5p; however, when we checked their abundance in miRbase[29], we unexpectedly

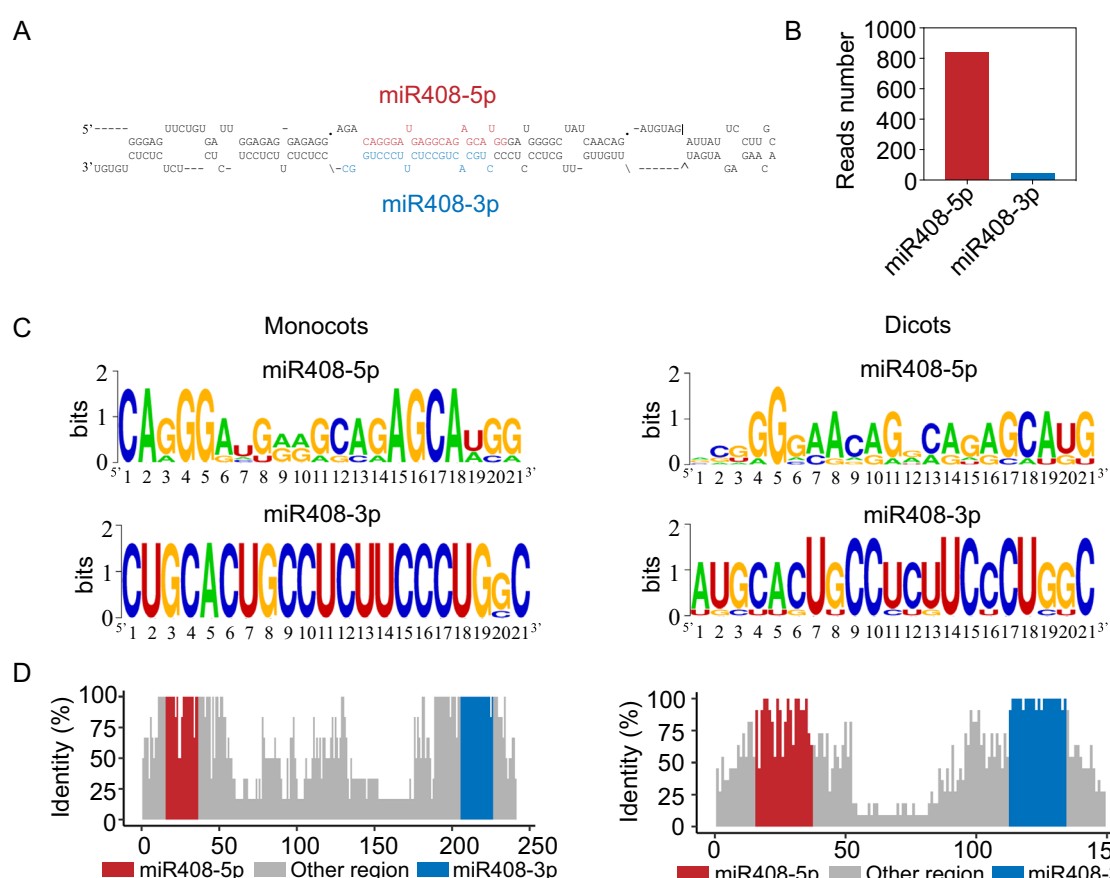

**Fig. 1 | Identification and conservation analysis of miR408-5p and miR408-3p in plants. A** Secondary structure of *MIR408* precursor in rice. Red sequences indicate mature miR408-5p while blue sequences indicate mature miR408-3p. **B** Reads number of rice miR408-5p and miR408-3p in miRbase. **C** Sequences of miR408-5p and miR408-3p in monocots and dicots shown by seqlogo. **D** The evolutionary analysis of *MIR408* precursor in monocots and dicots. The precursor sequence data are collected from miRbase website. Red and blue bars denote miR408-5p and miR408-3p regions, respectively. *X*-axis means the length of *MIR408* precursor and *y*-axis indicates the value of identical sequences. Source data underlying (**B**) are provided as a Source Data file.

found that the copy reads of miR408-5p were higher than those of miR408-3p in rice (Fig. 1B). We further calculated the normalized expression levels of miR408-5p and miR408-3p using PmiRExAt, a plant miRNA expression atlas database[30], and found a comparable quantity of miR408-5p to that of miR408-3p in specific tissues, such as leaf and callus, despite the overall lower accumulations of miR408-5p compared to miR408-3p in rice (Fig. S1A, B). Nevertheless, when we examined other rice miRNA precursors that can generate two mature miRNAs, we consistently observed significantly lower expression levels of non-conserved miRNAs than the conserved ones in all cases (Fig. S1A, B). These results suggest that miR408-5p is an abundant miRNA in rice and may therefore possess biological importance.

The presence of overlapping sequences between rice miR408-5p with miR408-3p* promoted us to explore the evolutionary history of these two miRNAs generated from miR408 precursor. The higher number of identical sequences in miR408-3p compared to miR408-5p suggested that miR408-3p is more conserved than miR408-5p (Fig. 1C, D). Interestingly, we found that both variations of miR408-5p and miR408-3p were a slightly higher in dicotyledons than in monocotyledons (Fig. 1C, D), implicating a potentially faster evolution of *MIR408* genes in dicots than in monocots. Moreover, we observed that the middle region of the *MIR408* stem-loop structure was significantly longer in monocots compared to dicots (Fig. 1D), and whether this difference contribute to the higher conversation of the mature miRNAs in monocots than that in dicots remains to be further determined.

## *IAA30* is targeted by miR408-5p in rice

To investigate the potential function of miR408-5p in rice, we predicted its target genes by PsRobot, a widely used small RNA analysis toolbox[31]. Interestingly, among the predicted targets using strict parameters, we identified an *Auxin-responsive Aux/IAA* gene family member, *IAA30*, with a potential target site located at the 3′Untranslated Region (UTR) (Fig. 2A). However, when we relaxed the computational prediction criteria, three *Aux/IAA* members, *IAA11*, *IAA19* and *IAA30*, were estimated as miR408-5p targets (Fig. S2A). Among them, *IAA30* had the highest complementarity to miR408-5p, while *IAA19* and *IAA11* harbored more bulges and mispairings with miR408-5p (Fig. S2A), implying that *IAA30* is more likely to be the genuine target of miR408-5p.

To experimentally test whether miR408-5p can regulate *IAA30* and *IAA19*, we performed a luciferase (LUC)-based reporter assay, in which *IAA30* or *IAA19* were transiently expressed in *Nicotiana benthamiana* leaves under the control of their own promoter in the presence of miR408-5p (Fig. 2B and S2B). On the one hand, the LUC signal was suppressed when both the wild-type *IAA30* 3′UTR and miR408-5p were introduced simultaneously, while it remained consistent when the mutated target sites of *IAA30* and miR408-5p were co-expressed (Fig. 2B, Fig. S3A, B). On the other hand, co-expression of *IAA19* and miR408-5p did not result in a profound reduction of LUC intensity (Fig. S2B). These results indicate that *IAA30* rather than *IAA19* is regulated by miR408-5p.

Given the fact that the involvement of IAA proteins in auxin signaling is conserved among monocots and dicots[32], we asked whether the targeting of *IAA* genes by miR408-5p is similarly conserved in different plants. To this end, we identified *IAA* orthologs in several representative monocots and dicots, and compared the sequence identities of their potential miR408-5p target sites, as well as other regions within *IAA30s'* 3′UTRs (Fig. S4A). We found the fragment of *IAA30s* that matches miR408-5p is more conserved in monocots than dicots (Fig. S4B), implicating that the orthologues of *IAA30s* in monocots are more likely to be targeted by miR408-5p compared to those in dicots. Nonetheless, when we surveyed the complementarity of miR408-5p with *IAA30* in six typical monocot species, we found that only *IAA30s* in rice and *Aegilops tauschii* showed potential targeting by

miR408-5p (alignment score ≤ 4) (Fig. S4C), indicating that the regulation of *IAA30* by miR408-5p in monocots has not undergone a strong selection pressure during evolution.

## miR408-5p is induced by auxin in rice

Since IAAs are involved in auxin signaling and miR408-5p targets *IAA30* in rice, we wondered whether miR408-5p was responsive to auxin treatment. Consistent with the expression data of rice *Aux/IAA* members reported before[33], our quantitative reverse transcription PCR (qRT–PCR) analysis showed that *IAA30* was significantly induced in rice roots by 10 μM indole-3-acetic acid (IAA) application (Fig. 2C). While the transcript of *MIR408* was enhanced 4-fold after 4 h IAA treatment, the induction of miR408-5p examined by a stem-loop qPCR was milder than *MIR408*, which was enhanced only 2-fold after 4 h IAA treatment and almost recovered after 8 h (Fig. 2D, E). Moreover, we detected that the accumulation of miR408-3p was also enhanced by auxin treatment (Fig. 2F). These results suggest that mature miR408-5p is subject to transcriptional and processing control in rice.

To further determine the regulation of *MIR408* and *IAA30* by auxin, we generated transgenic rice plants expressing the *GUS* reporter gene driven by the *MIR408* promoter (*MIR408p::GUS*) and *IAA30* promoter (*IAA30p::GUS*), respectively, and analyzed their performance before and after IAA treatment. GUS staining showed that *MIR408* was widely transcribed in rice plants, with particularly high expressions in roots, leaves, calli and leaf sheaths (Fig. S5). Moreover, we observed that the exogenous auxin treatment significantly enhanced the blue staining in both *MIR408p::GUS* and *IAA30p::GUS* transgenic plants (Fig. 2G, H). These results revealed that the primary transcripts, mature small RNA as well as the target of miR408-5p in rice are induced by auxin application.

## Over-expression of miR408-5p in rice

To gauge regulation of *IAA30* by miR408-5p in vivo, we introduced *MIR408* precursor under the *UBIQUITIN* (*UBI*) promoter into rice to generate *MIR408* overexpression transgenic plants (*MIR408-OE*). A dramatic increase in miR408-3p and a decrease of its targets *UCL7* and *UCL8* indicated that the pri-miR408 was indeed constitutively expressed and correctly processed in transgenic plants (Fig. S6A). As expected, the mature miR408-5p, as shown by stem-loop PCR, was significantly accumulated along with its precursor (Fig. S6B).

Since there was no developmental abnormality of roots existed in *MIR408-OE* plants compared with WT (Fig. S6C, D), we planted *MIR408-OE* and WT on fluid nutrient medium for 3 days and then treated them with 10 μM IAA for 11 days to determine the biological significance of *IAA30* by miR408-5p mediation. Unfortunately, we still could not find a noticeable difference between *MIR408-OE* and WT plants, although the status of primary roots and adventitious roots was easily observed to be altered by the application of exogenous auxin (Fig. S6E, F).

We were curious about why there was no morphological difference between *MIR408-OE* and WT plants after auxin treatment, because both miR408-5p and *IAA30* were substantially induced by auxin. To address this question, we first generated loss-of-function of *IAA30* mutants and observed their performances in rice (Fig. S7A). Similar to *MIR408-OE* plants, *iaa30* mutants lacked abnormal defects even when they were treated with auxin (Fig. S7B–E).

Given that there are 31 Aux/IAA genes in total in rice (Fig. S7F)[33], we speculated that the absence of root developmental differences between *MIR408-OE* and WT might be due to the redundancy of IAA30 with other IAAs. IAA11 was such a candidate according to its highest homology with IAA30 in rice (Fig. S7F), and therefore we generated *iaa30/iaa11* double mutants through the CRISPR-cas9 strategy (Fig. S7G). Nevertheless, the plant root morphologies in *iaa30/iaa11*, *MIR408-OE* and WT were still similar even though all of them were grown under 10 μM IAA treatment (Fig. S7H–K), implicating that not

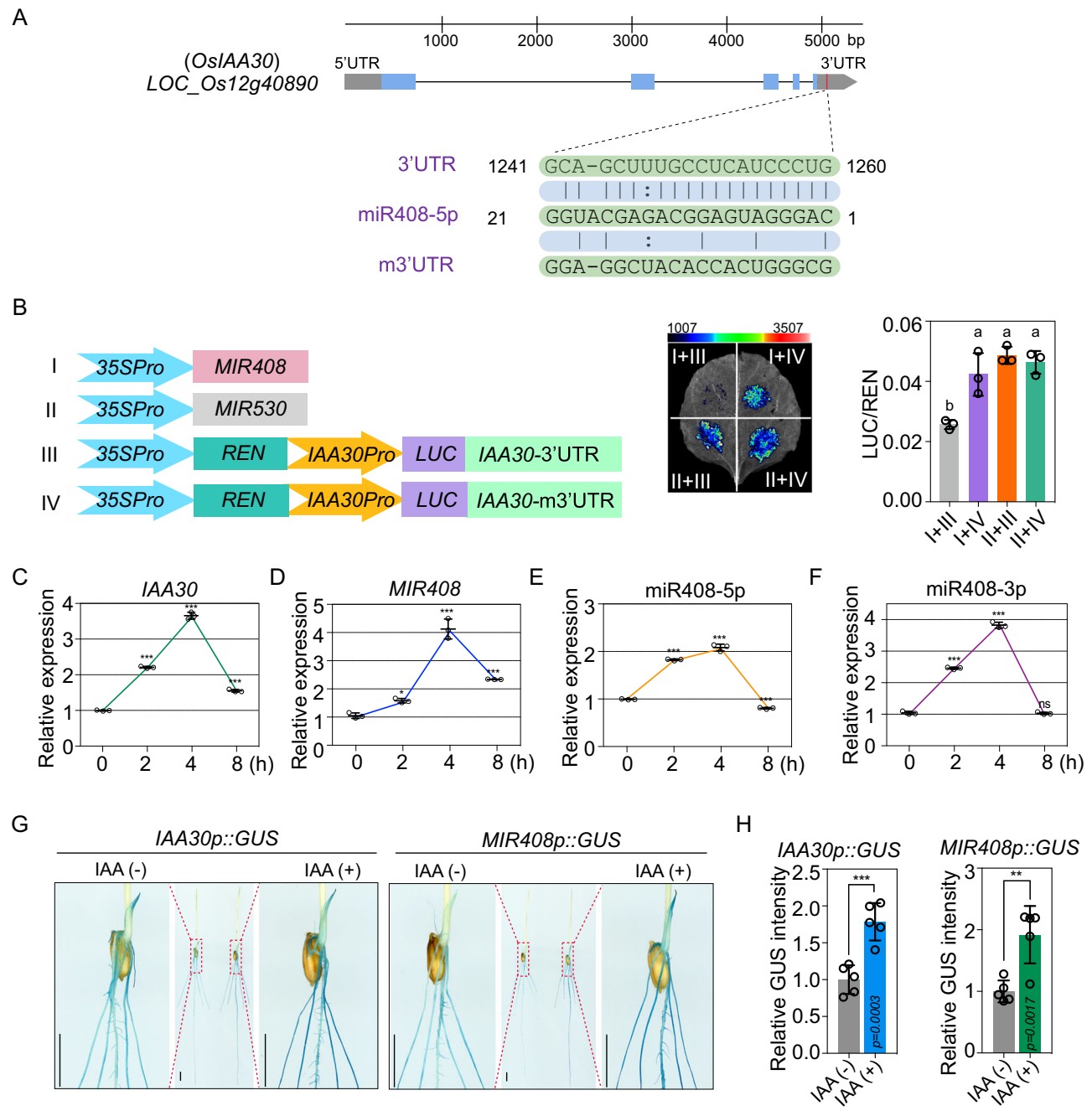

**Fig. 2 | miR408-5p targets *IAA30* and is induced by auxin in rice. A** Gene structure of *IAA30* and alignments of miR408-5p with target sites in *IAA30* 3'UTR and mutant 3'UTR (m3'UTR). **B** Validation of *IAA30* as miR408-5p target through transient expression analysis in *N. benthamiana* leaves. Left: The constructs in *A. tumefaciens* transiently introduced in *N. benthamiana* leaves. Middle: Representative photograph of firefly luciferase fluorescence signals when the indicated construct combinations were introduced in *N. benthamiana* leaves. Right: Relative reporter activity in *N. benthamiana* leaves expressing the indicated construct combinations. Error bars indicate SD (Tukey's honestly significant difference, $P < 0.05$). Quantitative reverse transcription PCR (qRT–PCR) analysis of *IAA30* (**C**), *MIR408* (**D**), miR408-5p (**E**) and miR408-3p (**F**) expression in rice with auxin treatment under different time. **G** *IAA30p-GUS* and *MIR408p-GUS* staining in 7-d-old rice plants with and without 4 h 10 μM IAA treatment. Bars = 1 cm. **H** The relative intensity of GUS before and after IAA treatment shown in (**F**). Error bars indicate SD (Student's *t*-test, \*\**P* < 0.01, \*\*\**P* < 0.001). Source data underlying (**B–F**, **H**) are provided as a Source Data file.

only IAA11 but also other IAAs may play redundant roles with IAA30 in auxin signaling in rice.

**miR408-5p is involved in auxin signaling**
Generally, IAA proteins play repressive roles in auxin signaling and may give rise to abnormal developments when they are over-accumulated in plants[34]. It has been shown that a gain-of-function mutation of *IAA11* influences lateral root development and altered auxin signaling in

rice[35], we asked whether a similar performance would happen if *IAA30* was overproduced. Therefore, we generated *IAA30* overexpression lines (*IAA30-OE*) and strikingly found that they had longer primary roots and shorter crown roots than WT (Fig. S8A, B), suggesting an important role of IAA30 in rice development, reminiscent that of IAA11 in rice[34,35].

To investigate the functional influence of miR408-5p on auxin signaling, we generated *MIR408* knock-out mutants via CRISPR-cas9

approach (Fig. S8C). Although no apparently developmental defects were observed under normal growth conditions (Fig. S8D, E); the primary roots in *mir408* mutants were significantly longer than those in WT when treated with 10 μM IAA (Fig. 3A, B). Moreover, although there were no obvious changes in the number of adventitious roots in *mir408* mutants compared with WT after IAA treatment, the three longest first-order adventitious roots in *mir408* were much shorter than in WT, similar to those in *IAA30-OE* lines (Fig. 3A, D). These results reflect that *mir408* mutants are insensitive to auxin treatment compared to WT.

Next, we attempted to determine the insensitivity of *mir408* mutants to auxin is due to the loss of miR408-5p or miR408-3p activity. To this end, we made small tandem target mimic (STTM) constructs and introduced them into rice plants to compromise either miR408-5p or miR408-3p activity. The expression levels of miR408-3p and miR408-5p in the *STTM-5p* and *STTM-3p* transgenic plants, as detected by stem-loop qPCR, were found to be similar to those in WT, suggesting that blocking of miR408-5p and miR408-3p activity by *STTM-5p* and *STTM-3p* was specific (Fig. S9A, B). Although there were no clear differences between *STTM-5p* and WT plants without an auxin treatment (Fig. S9C, D), we observed an interesting phenomenon after the application of 10 μM IAA. *STTM-5p* exhibited longer primary roots and shorter adventitious roots compared to WT, which resembled the phenotype observed in *mir408* mutants (Fig. 3E, F, and S9E–H). In contrast, this phenotype was absent in *STTM-3p* plants (Fig. S10), suggesting that miR408-5p is more likely than miR408-3p to be involved in auxin signaling in rice.

To further illustrate the involvement of *MIR408* in auxin signaling through miR408-5p-directed target regulation, we performed a transcriptomic analysis using WT, *STTM-5p, STTM-3p* and *IAA30-OE* plants with or without a 4 h 10 μM IAA treatment. In our RNA-seq dataset, the number of differentially expressed genes (DEGs) by auxin treatment was similar in WT and *STTM-3p*, higher than that in *STTM-5p* and *IAA30-OE* plants (Fig. S11A–E, Supplementary Data 1). Furthermore, the functional categories of DEGs in *STTM-5p* and *IAA30-OE* were more similar to each other, distinct from those in WT and *STTM-3p* (Fig. S11F, G and Supplementary Data 2), providing further evidences that *MIR408* regulates *IAA30* predominately through miR408-5p.

Finally, we determined the role of miR408-5p in auxin signaling dependent on *IAA30* via genetics. When we introduced *mir408* into *iaa11/30* background through crossing, we found the insensitivity to auxin treatment in *mir408* mutants was largely compromised (Fig. 3G, H, Fig. S12A, B). Simultaneously, we transformed *iaa11/30* mutants with *STTM-5p* constructs and obtained *STTM-5p/iaa11/30* transgenic plants. Root phenotypic observation showed that the long primary roots observed in *STTM-5p* plants after IAA treatment disappeared when they were in the *iaa11/30* background (Fig. 3I, J, Fig. S12C, D), suggesting that *IAA30* is genetically required for miR408-5p activity in auxin pathway.

Collectively, these data reveal that miR408-5p mediates auxin signaling transduction in rice through regulating *IAA30*.

## The regulatory machinery of IAA30 by miR408-5p

It has been generally considered that the regulation of target by miRNA in plants is predominantly through mRNA cleavage[36]. miR408-5p seems to be analogous in this case, since overexpression of miR408-5p led to a decrease in *IAA30* transcripts (Fig. 4A).

However, when we determined *IAA30* expression by qRT-PCR in *mir408* mutants, we were unable to detect any increase in *IAA30* mRNA compared with that in WT (Fig. 4A), although we readily detected an enhanced expression of *UCL7* and *UCL8*, two well-known targets of miR408-3p (Fig. S13A). Furthermore, compared to WT, we did not observe an increase in *IAA30* transcripts in *STTM-5p* plants (Fig. 4A), whereas a conspicuous increase in the expression of *UCL7* and *UCL8* was detected in *STTM-3p* lines (Fig. S13B), implying that the regulatory

manner of miR408-5p on its target may differ from that of miR408-3p. Additionally, the expression of *IAA30* is comparable in WT and *STTM-3p* plants (Fig. S13C), indicating that miR408-3p may not be involved in *IAA30* regulation in rice.

When we conducted a 5′rapid amplification of complementary DNA (cDNA) ends (5′RACE) analysis to determine cleavage sites of *IAA30* mRNA by miR408-5p, we were unable to detect any cleavage events on *IAA30* mRNA. Moreover, despite the effectiveness of rice degradome databases in identification of rice mRNA targets, we did not detect any noteworthy cleavage events in the complementary region between miR408-5p and *IAA30* (Fig. S13D). These results indicated a non-canonical mode of action for miR408-5p in target regulation.

The reduction of *IAA30* expression in *MIR408-OE* plants coupled with the absence of enhanced *IAA30* mRNAs in *mir408* and *STTM-5p* plants led us to speculate that miR408-5p may change regulatory mechanism of *IAA30* when its abundance is altered. To test this possibility, we first performed a degradome sequencing to examine *IAA30* slicing events in *MIR408-OE* plants. Strikingly, *IAA30* was cleaved by miR408-5p in the base-pairing region, although the major transcript ends deviated approximately 1 nucleotide upstream of the conventional cleavage site (between 9th and 10th relative to the 5′end of miRNA) (Fig. 4B, C). 5′RACE experiment confirmed this cleavage event in *MIR408-OE* transgenic plants (Fig. S14A). Additionally, a similar cleavage event was also detected in another degradome library from rice explants induced from major roots by a synthetic auxin, 2,4-dichlorophenoxyacetic acid (2,4-D) (Fig. 4C, D).

In addition, as miR408-5p is more abundant in rice callus than other tissues (Figs. S1B, S5), we constructed the third degradome library from 14-day-old rice callus that was generated from mature embryos under high auxin conditions. Interestingly, a consistent cleavage site of *IAA30* was detected between the 8th and 9th nucleotide of miR408-5p, similar to that observed in *MIR408-OE* plants (Fig. 4C, E). Together, these results suggest that miR408-5p downregulates *IAA30* by mRNA cleavage when it is artificially over-generated or accumulates above a basal threshold under high-auxin environments.

Considering the difficulty in obtaining a specific antibody of IAA protein due to the high similarity of amino acids among different family members, we introduced a Flag-tagged IAA30 with WT 3′UTR (35S-Flag-IAA30-UTR) or mutated 3′UTR (35S-Flag-IAA30-mUTR) into rice protoplast to determine the protein accumulation in miR408-5p gain- and loss-of-function plants (Fig. 4F). As anticipated, both the transcripts and proteins of *IAA30* were reduced in *MIR408-OE* rice protoplasts (Fig. 4G, Fig. S14B). However, the protein abundances but not the mRNA levels of *IAA30* were increased in the protoplasts of *mir408* mutants (Fig. 4H, Fig. S14B). Furthermore, this increase in IAA30 protein was no longer observed when we introduced a mutated target site into *IAA30* (Fig. 4H, Fig. S14B).

Next, we transiently expressed *Flag-IAA30* in *STTM-5p* line protoplasts, and found that the protein, but not the mRNA, of *IAA30* was higher in *STTM-5p* than in WT protoplasts (Fig. 4I, Fig. S14B). By contrast, when we introduced 35S-Flag-IAA30-mUTR, the protein was accumulated similarly in protoplasts from WT and *STTM-5p* lines (Fig. 4I, Fig. S14B). Moreover, when we used *STTM-3p* plants to generate protoplasts and introduced the same constructs, we found that neither mRNA nor protein of *IAA30* was significantly enhanced in *STTM-3p* protoplasts compared with WT (Fig. 4J, Fig. S14B). These results suggest that the translation of IAA30 is specially reinforced when miR408-5p activity is compromised.

To further demonstrate the protein inhibition as the regulatory manner of *IAA30* by miR408-5p under normal conditions, we generated transgenic rice expressing Flag-tagged IAA30 with either the WT 3′UTR or a mutated 3′UTR driven by the native *IAA30* promoter (named *IAA30* and *IAA30m* lines, respectively) (Fig. 4K). Compared

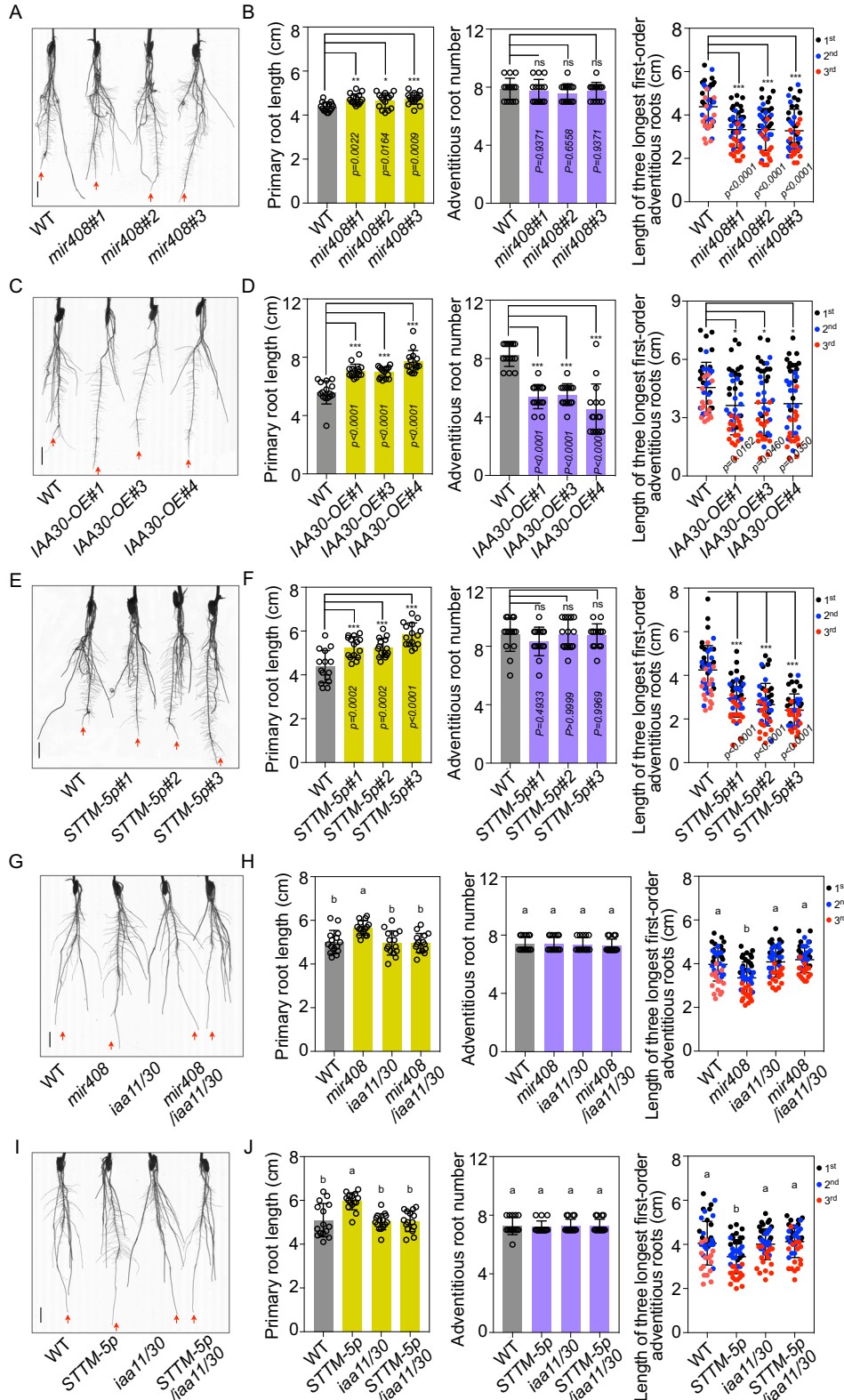

**Fig. 3 | miR408-5p is involved in auxin signaling through regulation of *IAA30*.** Representative photographs of 2-week-old *mir408* (**A**), *IAA30-OE* (**C**), *STTM-5p* (**E**), *mir408/iaa11/30* (**G**), *STTM-5p/iaa11/30* (**I**) and respective WT plants under 10 μM IAA treatment. The red arrow indicates the primary root. Bar = 1 cm. **B**, **D**, **F**, **H**, **J** Primary root length, adventitious root number and the length of three longest first-order crown roots in 2-week-old WT and the indicated plants shown in (**A**), (**C**), (**E**), (**G**) and (**I**) under 10 μM IAA treatment. Error bars indicate SD (Multiple comparisons test, *P < 0.05; **P < 0.01; ***P < 0.001; ns not significant; Tukey's honestly significant difference, *P* < 0.05). Source data underlying (**B**, **D**, **F**, **H**, **J**) are provided as a Source Data file.

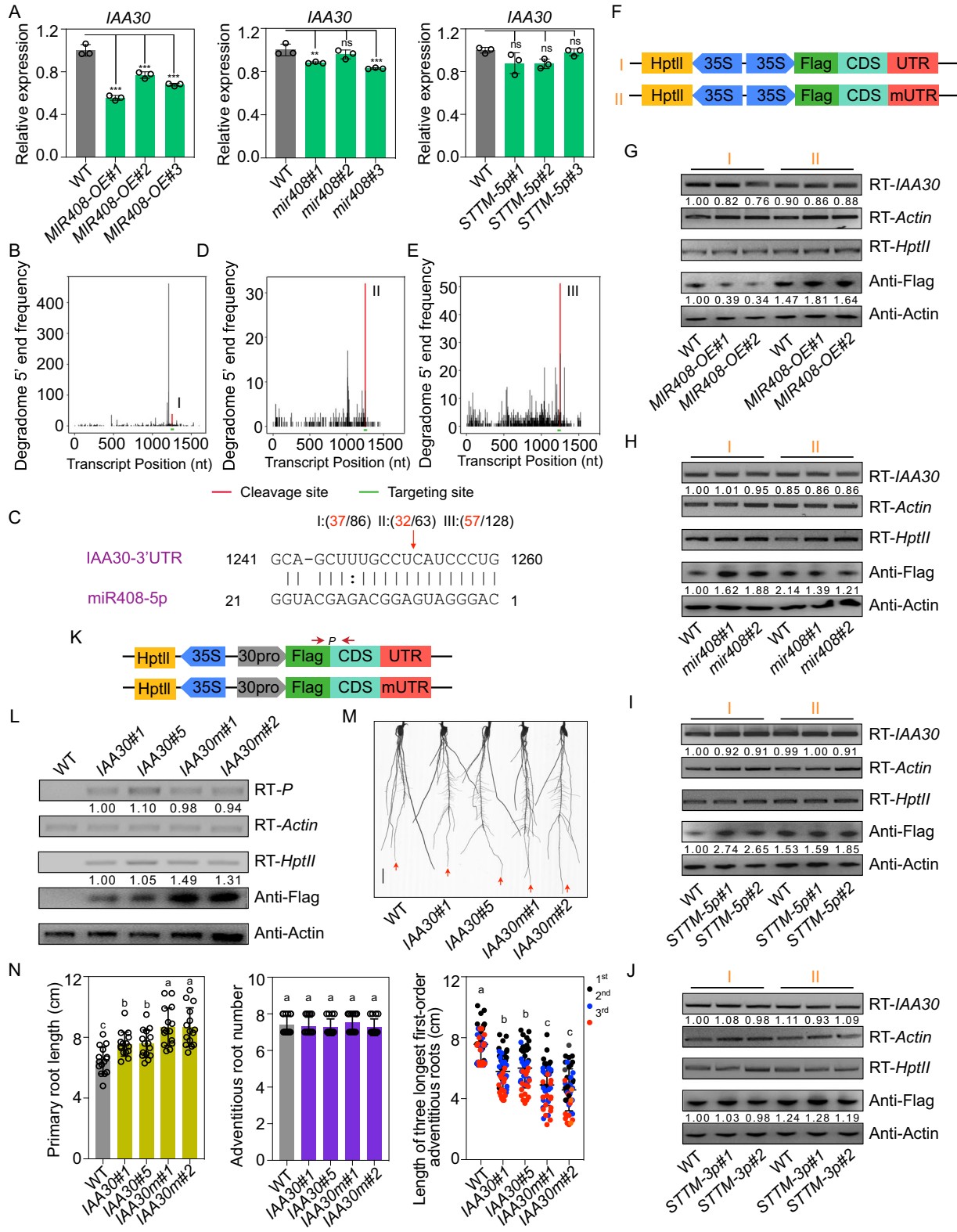

with *IAA30* lines, although the mRNA levels were detected similarly, the protein of Flag-IAA30 was significantly higher in the *IAA30m* lines (Fig. 4L), which result in longer primary roots and shorter adventitious roots, especially under the conditions with auxin treatment (Fig. 4M, N, Fig. S15A, B).

Taken together, our finding shows that miR408-5p mediates the regulation of *IAA30* through translation repression under normal conditions, but switches to an mRNA digestion fashion if it is over-produced or enhanced by auxin.

## miR408-5p is regulated by miR156 and IPA1 in auxin signaling

In *Arabidopsis thaliana*, miR408 is positively modulated by an SPL family protein, SPL7, to mediate the metabolism and development in response to copper and light[37]. We asked whether the precursor and

**Fig. 4 | Regulation manners of miR408-5p to *IAA30* in rice. A** Expressions of *IAA30* in *MIR408-OE*, *mir408* and *STTM-5p* plants. Error bars indicate SD (Multiple comparisons test, **P < 0.01; ***P < 0.001; ns, not significant). Cleavage events of miR408-5p to *IAA30* mRNA shown by degradome sequencing data obtained from *MIR408-OE* transgenic plants (**B**), explants from major root under 3d 10 µM 2,4-D treatment (**D**) and 14d rice callus induced by mature embryos (**E**). The red vertical bar indicates the positions of reads with the highest frequency mapped to the miR408-5p binding site. The green horizontal bar denotes miR408-5p targeting region. **C** Base pairing and the highest cleavage frequency of *IAA30* 3'UTR by miR408-5p observed in the indicated experiments shown in (**B**), (**D**), (**E**). The frequency of the sequenced 5'ends is plotted against the position in the *IAA30* target site. **F** The constructs that were introduced into rice protoplast to determine the effect of miR408-5p on *IAA30*. HptII, *Hygromycin phosphotransferase II*. RNA and protein abundance of *IAA30* when normal *IAA30* 3'UTR (I) or mutated *IAA30* 3'UTR (*mUTR*) (II) construct shown in (**F**) was introduced into the protoplast from *MIR408-OE* (**G**), *mir408* (**H**), *STTM-5p* (I) and *STTM-3p* (**J**) plants. The relative transcript and protein level of *IAA30* were calculated using *Actin* as an endogenous control and were set to 1 in WT protoplast transformed with normal *IAA30* 3'UTR. RT-*HptI* was used to show the efficiency of plasmid transiently introduced into the indicated protoplast. Two independent transgenic lines were used to make protoplast. RT indicates transcript abundance determined by reverse transcription PCR; Anti denotes protein level examined by Western blot analysis using the indicated antibody. The number below each lane represents the relative amount compared with that in WT protoplast transformed with normal *IAA30* 3'UTR. **K** The constructs used to generate transgenic rice that was introduced Flag-tagged IAA30 with WT 3'UTR (*IAA30*) or mutated 3'UTR (*IAA30m*) under control of the *IAA30* native promoter. The red arrows with letter P mark the RT-PCR region for RNA determination. **L** RNA and protein abundance of *IAA30* in WT and transgenic plants with indicated constructs shown in (**K**). The number below the lane represents the relative amounts of transcripts and proteins. **M** Representative photograph of 2-week-old WT, *IAA30* and *IAA30m* transgenic plants under 10 µM IAA treatment. The red arrow indicates the primary root. Bar = 1 cm. **N** Primary root length, adventitious root number and the length of three longest first-order crown roots in 2-week-old WT, *IAA30* and *IAA30m* transgenic plants under 10 µM IAA treatment. The different letters on top of each bar denote significant differences (Tukey's honestly significant difference, *P* < 0.05). Source data underlying (**A**, **G**–**J**, **L**, **N**) are provided as a Source Data file.

mature miR408-5p could be regulated by SPL proteins in auxin pathway in rice. To this end, we first searched for GTAC-motifs, the potential binding elements of SPLs, in 1.5 kb promoter upstream of rice *MIR408*. Like in *A. thaliana*, there were several GTAC-motifs in the *MIR408* promoter, especially enriched within the 600 bp proximal region (Fig. 5A). In rice, the gene *IPA1*, which encodes SPL14, is a key regulator of architecture plasticity and has been considered as a new "Green Revolution" gene[20,23]. We examined the published chromatin immunoprecipitation (ChIP)-seq dataset of IPA1[38], and found a significant enrichment of IPA1 at *MIR408* promoter, particularly in the proximal region (Fig. 5A), suggesting that *MIR408* is an authentic target of IPA1 and works downstream of IPA1 in the regulatory pathway. Yeast One-Hybrid (Y1H) analysis and ChIP-qPCR validated IPA1 as a plausible upstream regulator of *MIR408* through interacting with its promoter proximal fragments (Fig. 5B, C).

To further confirm the effect of IPA1 on *MIR408* expression, we made *35S::GUS* and *35S::IPA1* effector constructs, and co-infiltrated them into *N. benthamiana* leaf epidermal cells along with a reporter system consisting of *MIR408* promoter-driven LUC (*MIR408p::LUC*) and REN (Fig. 5D). As shown in Fig. 5E, co-expression of *MIR408p::LUC* with IPA1 robustly enhanced the LUC activity, indicating that IPA1 positively regulates *MIR408*.

To trace the function of IPA1 to *MIR408* in *planta*, we checked the expression of precursor and mature miR408-5p in *IPA1* over-expression (*IPA1-OE*) and loss-of-function rice plants. As shown in Fig. 5F, *MIR408* gene as well as miR408-5p were highly accumulated in *IPA1-OE*, while dramatically suppressed in *ipa1* mutants and miR156 overproducing lines, in which *IPA1* was silenced[39].

Since further expression analysis showed that mature miR156 was repressed whereas *IPA1* was induced by IAA treatment (Fig. 5G), we next gauged whether *MIR156-OE* plants and *ipa1* mutants displayed insensitivity to auxin application. Compared with WT, both *ipa1* mutant and *MIR156-OE* plants exhibited long seminal roots and short crown roots, similar to those in *mir408* mutants and *STTM-5p* plants (Fig. 5H, I). These results suggest that miR156 may inhibit the activity of miR408-5p in mediating auxin signaling through targeting *IPA1*. It is worth noting that the number of adventitious roots under IAA treatment was lower in *MIR156-OE*, but comparable in *ipa1*, *mir408* and *STTM-5p* plants (Fig. 5I), suggesting the decrease in adventitious root number caused by miR156 over-accumulation may be dependent on repression of other SPLs rather than IPA1 and miR408-5p.

To determine the genetic relationship between *MIR408* and *MIR156*, we introduced *MIR408-OE* constructs into *MIR156-OE* plants by transformation, and noticed that the long primary roots and short adventitious roots in *MIR156-OE* compared to WT was partially restored when *MIR408* was overexpressed (Fig. 5J, K). Meanwhile,

when we crossed *ipa1* mutants with *MIR408-OE* plants, we found that the long seminal roots and short crown roots in *ipa1* were significantly attenuated when *MIR408-OE* was introduced (Fig. 5L, M). These data together indicate a genetic epistasis of miR408-5p to miR156 and IPA1 in rice auxin signaling.

Overall, these results characterize a regulatory cascade in rice auxin signaling that is constituted by miR156-*IPA1*-miR408-5p-*IAA30*.

## miR156-*IPA1*-miR408-5p-*IAA30* circuit regulates leaf inclination in rice

As we have demonstrated the regulation of *IAA30* by miR408-5p in auxin signaling, we subsequently wondered about the biological significance of this module in rice.

Leaf angle is an important agronomic trait of crops that influences photosynthetic efficiency and yield[40]. Several studies have implicated the significance of auxin in regulation of leaf inclination[41-43]. In our *IAA30-OE* transgenic lines, we found an enlarged leaf inclination compared with WT (Fig. S16A–C). The paraffin section revealed that the cell width of the fourth layer parenchyma cells at the adaxial side of the lamina joint of flag leaves was significantly increased in *IAA30-OE* compared to WT (Fig. S16D, E), consistent with the previous report that auxin-mediated leaf inclination regulation predominantly depends on the change of cell size rather than cell number[41].

By phenotypic observation, we found that both *mir408* mutants and *STTM-5p* plants showed increased leaf angles after heading (Fig. 6A–F), although there was no obvious reduction of leaf inclination in our *MIR408-OE* plants (Fig. S16F–J). Most flag leaves on major stems with panicles in *mir408* mutants and *STTM-5p* plants were above 60°, in contrast with those below 30° in WT (Fig. 6B, E). When we calculated the ratio of inclination of leaves≥90° in all stems harboring panicles, we observed that flag leaves over 30% in *mir408* mutants and 20% in *STTM-5p* plants were more than 90°, while this ratio was below 5% in WT (Fig. 6C, F). Meanwhile, we did not observe a distinctive difference in leaf angles between *STTM-3p* and WT plants (Fig. 6G–I). In line with these phenotypic observations, the longitudinal sections displayed an increased cell width of the adaxial side of lamina joint in *mir408* mutants and *STTM-5p* plants, while no alteration was found in WT and *STTM-3p* lines (Fig. 6J, K). Thus, the presence and the absence of enlarged leaf inclination in *STTM-5p* and *STTM-3p* plants, respectively, defined that regulation of leaf inclination by *MIR408* is mainly through miR408-5p.

Additionally, compared with WT, very few leaves in *IAA30* plants were observed that enhanced their angles, while those were readily found in *IAA30m* lines (Fig. S16K–M), further indicating that the miR408-5p-IAA30 regulon mediates leaf development in rice. We could not observe a visible change in leaf angle in *MIR408-OE* plants

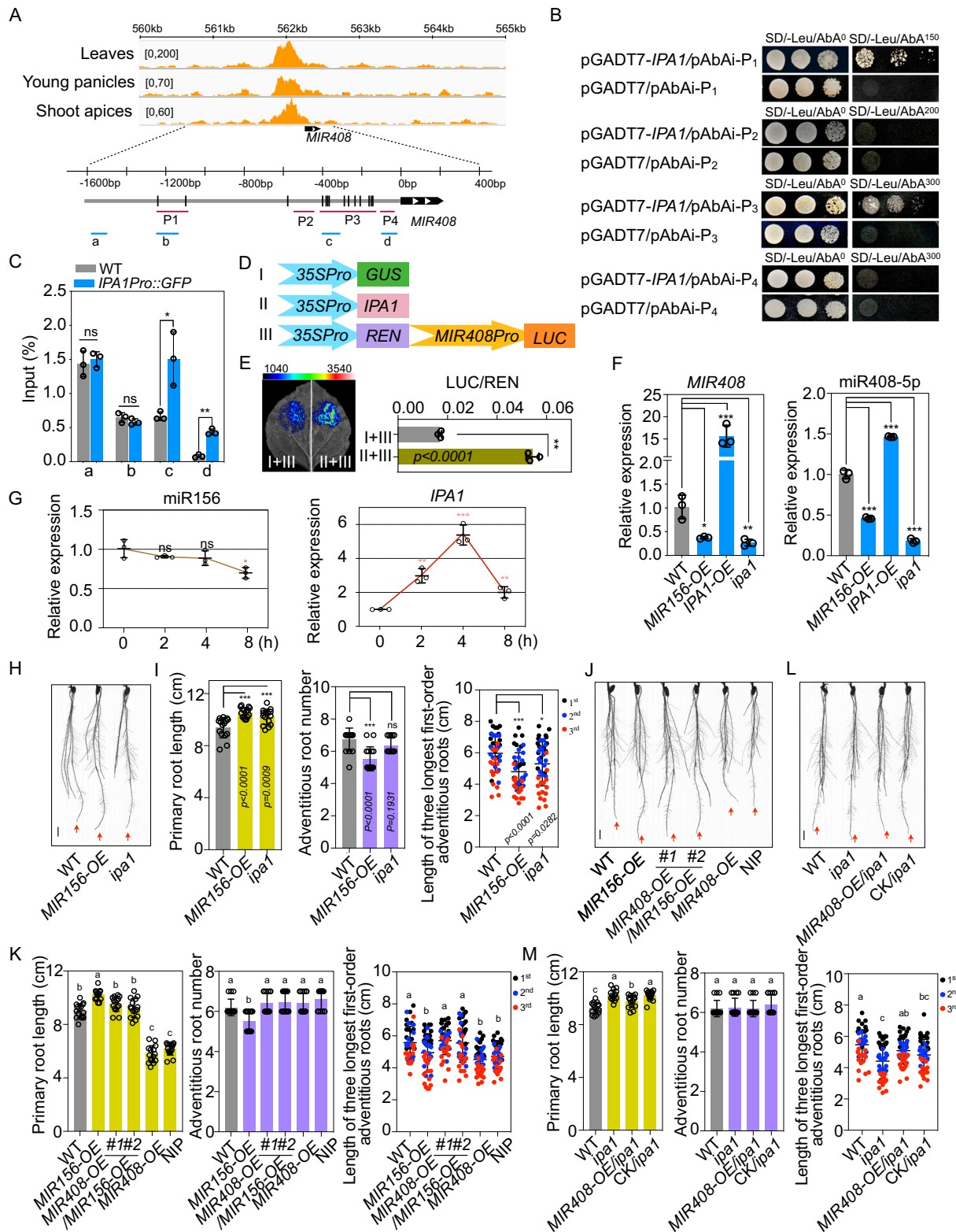

(Fig. S16F–J), possibly because only a decrease of *IAA30* cannot affect leaf inclination due to the heavy redundancy of IAA family proteins.

Because IPA1 promotes miR408-5p accumulation through associating with *MIR408* cis-elements, IPA1 likely represses leaf inclination. Indeed, we found that the flag leaf angles in *miR156-OE* plants and *ipa1* mutants, in which miR408-5p accumulations were decreased, were almost 100°, much higher than that of WT (Fig. 6L–N). Moreover,

cytological analysis showed that the width of the fourth layer cells of the lamina joint at adaxial side was substantially increased in *miR156-OE* and *ipa1* compared to WT (Fig. 6O, P), further implicating their enlarged leaf inclinations possible because of the miR408-5p deficiency. In addition, the enlarged leaf inclination in *miR156-OE* and *ipa1* was remarkably compromised when overexpression of *MIR408* was introduced, indicating that regulation of leaf angle development by

**Fig. 5 | IPA1 regulated by miR156 acts upstream of miR408-5p in auxin signaling. A** The association of IPA1 with *MIR408 cis*-elements in IPA1 ChIP-Seq datasets from three different rice tissues. The black bar on the promoter of *MIR408* shown in the picture represents GTAC-motifs. P1 to P4 marked in red color denotes the fragments that were used to construct vectors for Yeast One-Hybrid (Y1H) analysis and a to d marked by blue color denotes the fragments that were examined by ChIP-qPCR. **B** Y1H analysis for the interaction of IPA1 with *MIR408* promoter. Yeast cells growing on selective media without Leu and with the indicated concentration of Aureobasidin A (AbA) represent positive interactions. **C** ChIP-qPCR determination of IPA1 and *MIR408* promoter interactions. GFP antibody was used for immunoprecipitation and *Actin* was used as an endogenous control to calculate the relative enrichment of IPA1 to different fragments of *MIR408* promoter. **D** The reporter and effector constructs used for transient expression in *N. benthamiana* leaves for analysis of IPA1 activity to *MIR408* activation. **E** Examination of IPA1 inductive activity to *MIR408* transcription in *N. benthamiana* leaves. Left: Representative photograph of firefly luciferase fluorescence signals when the indicated constructs were introduced in *N. benthamiana* leaves. Right: Relative luciferase fluorescence signals in *N. benthamiana* leaves expressing the indicated reporters and effectors. Error bars indicate SD (Student's *t*-test, **$P < 0.01$). **F** qRT-PCR analysis of *MIR408* and mature miR408-5p expressions in WT, *MIR156-OE*, *IPA-OE* and *ipa1* rice plants. Error bars represent SD for three replicates. **G** Mature miR156 and

*IPA1* expressions in rice under 10 μM IAA treatment with the indicated time. **H** Representative phenotype of seedling roots of 2-week-old WT (ZH11), *MIR156-OE* transgenic plants and *ipa1* mutants under 10 μM IAA treatment. The red arrow indicates primary roots. Bar = 1 cm. **I** Primary root length, adventitious root number and the length of three longest first-order crown roots in 2-week-old WT (ZH11), *ipa1* mutants and *MIR156-OE* transgenic plants under 10 μM IAA treatment. Error bars indicate SD (Multiple comparisons test, **$P < 0.01$; ***$P < 0.001$; ns, not significant). **J** Representative phenotype of seedling roots of 2-week-old WT (ZH11), *MIR156-OE*, *MIR408-OE/MIR156-OE*, *MIR408-OE* and NIP plants under 10 μM IAA treatment. The red arrow indicates primary roots. Bar = 1 cm. **K** Primary root length, adventitious root number and the length of three longest first-order crown roots in 2-week-old WT (ZH11), *MIR156-OE, MIR408-OE/MIR156-OE, MIR408-OE* and NIP plants under 10 μM IAA treatment. Error bars indicate SD (Tukey's honestly significant difference, $P < 0.05$). **L** Representative phenotype of seedling roots of 2-week-old WT (ZH11), *ipa1, MIR408-OE/ipa1* and a negative plant used for crossing without *MIR408-OE* (CK/*ipa1*) under 10 μM IAA treatment. The red arrow indicates primary roots. Bar = 1 cm. **M** Primary root length, adventitious root number and the length of three longest first-order crown roots in 2-week-old WT (ZH11), *ipa1, MIR408-OE/ipa1* and CK/*ipa1* plants under 10 μM IAA treatment. Error bars indicate SD (Tukey's honestly significant difference, $P < 0.05$). Source data underlying (**C**, **E**−**G**, **I**, **K**, **M**) are provided as a Source Data file.

miR156 and IPA1 in rice is genetically dependent on the potential role of miR408-5p (Fig. 6Q–V).

Taken together, our data illustrate the biological significance of the modulatory circuit formed by miR156-*IPA1*-miR408-5p-*IAA30* in auxin signaling in rice.

### The effects of miR393-*TIR1/AFB* module on miR156-*IPA1*-miR408-5p-*IAA30* circuit in rice

Because auxin signaling primarily begins with the sensing of auxin molecules by TIR1/AFBs F-box receptor proteins, we pursued whether the response of miR408-5p to auxin was also dependent on these receptors. For this purpose, we examined the expression of *MIR408* and miR408-5p accumulations in two miR393 overexpression rice lines (*MIR393a-OE* and *MIR393b-OE*), in which major *TIR1/AFBs* genes were silenced[44]. Intriguingly, we observed that both *MIR408* and miR408-5p showed reduced responsiveness to IAA application in miR393 overproducers compared to WT (Fig. S17A, B), which is consistent with the results obtained from *tir1* mutants (Fig. S17C, D). These data collectively show that auxin receptors are essential for the induction of miR408-5p by auxin. In accordance with this, we also observed a greater bending of the lamina joint in *miR393a-OE* plants (Fig. S18A–E), similar to those in *mir408* and *STTM-5p*.

Next, we investigated whether miR393-TIR1/AFBs module influenced the expressions of miR156 and *IPA1*, the upstream regulators of miR408-5p. We found that both the basal expression and induction of *IPA1* by auxin were dramatically compromised in *MIR393a-OE* and *MIR393b-OE* plants (Fig. S19A), suggesting that auxin reception by TIR1/AFBs is important for *IPA1* regulation. However, when we measured the abundance of mature miR156 in *MIR393a-OE* and *MIR393b-OE* plants, we found no difference before auxin treatment compared with WT (Fig. S19B). Nonetheless, when we detected the repression of miR156 after auxin treatment, we found it was significantly enhanced by overexpression of miR393 (Fig. S19B), indicating that the response to high auxin environment, but not the basal expression of miR156 in rice, is affected by the miR393-TIR1/AFBs module.

## Discussion

Here, we present a regulatory network in rice involving three miRNAs that mediates plant architecture through auxin signaling (Fig. 7). Most surprisingly, we find that miR408-5p controls *IAA30* through translational repression under normal conditions, while shifts to mRNA degradation to silence it when miR408-5p is induced by high auxin perception. To our knowledge, this is the first study that clearly

elucidate the circumstance under which miRNA can switch regulatory modes in target regulation.

### miRNA-miRNA interaction as a master regulatory circuit in plant development

Indeed, the versatile roles of miRNA-target modules in plants are being gradually uncovered. The verified targets of plant miRNAs encompass a diverse range of modulatory proteins, especially transcription factors. This implies that miRNA-mediated targets may constitute a regulatory circuit through modulating another miRNA and its targets, allowing for hierarchical and efficient participation in plant development or adaptation.

For example, miR156 represses miR172 through suppressing *SPL3*, which to release a class of APETALA 2 transcription factors that control growth transition and reproductive development[45]. In leaf morphogenesis, miR164 targets *CUP-SHAPED COTYLEDON 2* mRNA, which is decoyed by miR319-TEOSINTE BRANCHED 3, a key regulon involved in a gradual increase of leaf complexity in the newly formed organs[46]. Due to the high conservation of these miRNAs and targets among diverse plants, such regulatory cascades are vital in plant growth and have undergone strong selective pressure during evolution.

We find herein that IPA1, an important contributor to the desirable architecture trait in rice, promotes miR408-5p accumulation through binding *MIR408 cis*-elements. As miR156 and miR393 regulate *IPA1*, the hierarchy interaction between these miRNAs is critical to regulate leaf lamina joint development in rice. Interestingly, the repression of miR156 by exogenous auxin treatment is exacerbated rather than attenuated when the canonical auxin receptors *TIR1/AFBs* are inhibited, implicating that regulation of miR156 by auxin may rely on alternative auxin receptors, such as Auxin-binding protein 1 (ABP1) as illustrated recently[47], or be subject to feed-back regulations by other components in the auxin signaling pathway.

### Contribution of translation repression and mRNA decay in silencing target by a miRNA

During the last two decades, numerous studies revealed two modes of negative regulation by miRNA on targets: translation repression and mRNA cleavage. The degree of complementarity between the miRNA and its target has been generally considered as a key determinant of the mechanism used, such that perfect match facilitates cleavage, while less pairing promotes to suppress protein translation[8]. In animals, most miRNAs display less matches to their target mRNAs than in plants, so translation inhibition is considered as

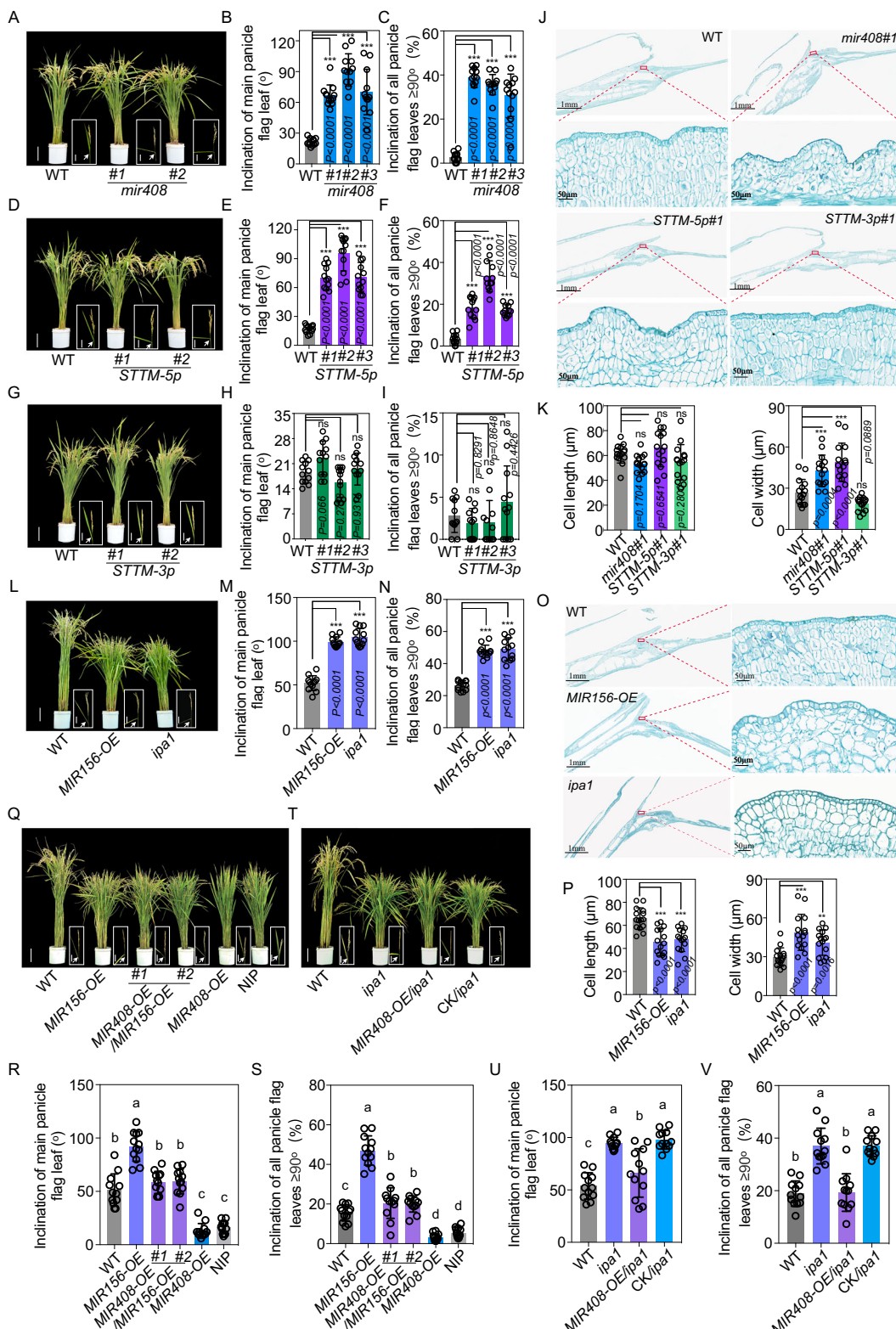

the critical activity of miRNAs in animals, preceding target mRNA decay[14].

Unlike animal miRNA, plant miRNA matches nicely with targets, and thereby often presents as a guide for target RNA cleavage. Another interpretation of limited events of translation inhibition in plants is that plants lack GW182 homolog proteins, which are the core components for miRNA-mediated translational repression in animals[14].

However, several conserved miRNAs, including miR156, miR172, miR398, miR164 and miR165/166, have been reported that translation repression is engaged in target silencing, independent of slicing the same gene transcripts[3,5,6,10]. Despite a deficiency in miRNA-mediated translational inhibition, several mutants in plants, such as *katanin1* (*knt1*) and *suo1*, display only mild abnormality in development, whereas others, including *ago1-27*, *altered meristem program 1* (*amp1*)

**Fig. 6 | miR156-*IPA1*-miR408-5p-*IAA30* circuit regulates rice leaf inclination.** Phenotypic observations of *mir408* (**A**), *STTM-5p* (**D**) and *STTM-3p* (**G**) and respective WT plants (Bars = 10 cm). Boxed regions are enlarged to the performance of flag leaves with panicles (Bars = 1 cm). White arrows indicate the lamina joints. Inclination of flag leaves on the main stem with panicles in *mir408* (**B**), *STTM-5p* (**E**) and *STTM-3p* (**H**) and respective WT plants. Angles of flag leaf at 40 days after heading were measured and data are presented as means ± SD (standard deviation, *n* = 12). Statistical analysis was conducted by using Multiple comparisons test (\*\*\**P* < 0.001; ns not significant). The ratio of flag leaves with angle≥90°from all flag leaves on stems with panicles in *mir408* (**C**), *STTM-5p* (**F**) and *STTM-3p* (**I**) and respective WT plants. Statistical analysis was executed by using Multiple comparisons test (\*\*\**P* < 0.001; ns not significant). **J** Longitudinal section of the adaxial region of the lamina joint in WT, *mir408*, *STTM-5p* and *STTM-3p* flag leaves at 40 days after heading. The marked regions in red color were magnified to highlight the differences. **K** Cell length and cell width of adaxial parenchyma cells of lamina joint in WT, *mir408*, *STTM-5p* and *STTM-3p* plants. The fourth layer parenchyma cells at the adaxial side were calculated and statistically analyzed by Multiple comparisons test (\*\*\**P* < 0.001; ns not significant). **L** Typical plant phenotypes of

WT (ZH11), *miR156-OE* transgenic plants and *ipa1* mutants (Bars = 10 cm). Boxed regions are enlarged to the performance of flag leaves with panicles (Bars = 1 cm). White arrows indicate the lamina joints. **M** Inclination of flag leaf on the main stem with panicle in WT (ZH11), *miR156-OE* and *ipa1* plants. **N** The ratio of flag leaves with angle≥90°from all flag leaves on stems with panicles in WT (ZH11), *miR156-OE* and *ipa1* plants. **O** Longitudinal section of the adaxial region of the lamina joint in WT (ZH11), *miR156-OE* and *ipa1* flag leaves at 40 days after heading. **P** Cell length and cell width of adaxial parenchyma cells of lamina joint in WT (ZH11), *miR156-OE* and *ipa1* plants. Phenotypic comparisons in WT (ZH11), *miR156-OE*, *miR408-OE/miR156-OE*, *miR408-OE*, NIP plants (**Q**) and in WT (ZH11), *ipa1*, *MIR408-OE/ipa1* and CK/*ipa1* plants (**T**). Inclination of flag leaf on the main stem with panicle in WT (ZH11), *miR156-OE*, *miR408-OE/miR156-OE*, *miR408-OE*, NIP plants (**R**) and in WT (ZH11), *ipa1*, *MIR408-OE/ipa1* and CK/*ipa1* plants (**U**). The ratio of flag leaves with angle ≥90° from all flag leaves on stems with panicles in WT (ZH11), *miR156-OE*, *miR408-OE/miR156-OE*, *miR408-OE*, NIP plants (**S**) and in WT (ZH11), *ipa1*, *MIR408-OE/ipa1* and CK/*ipa1* plants (**V**). Source data underlying (**B**, **C**, **E**, **F**, **H**, **I**, **K**, **M**, **N**, **P**, **R**, **S**, **U** and **V**) are provided as a Source Data file.

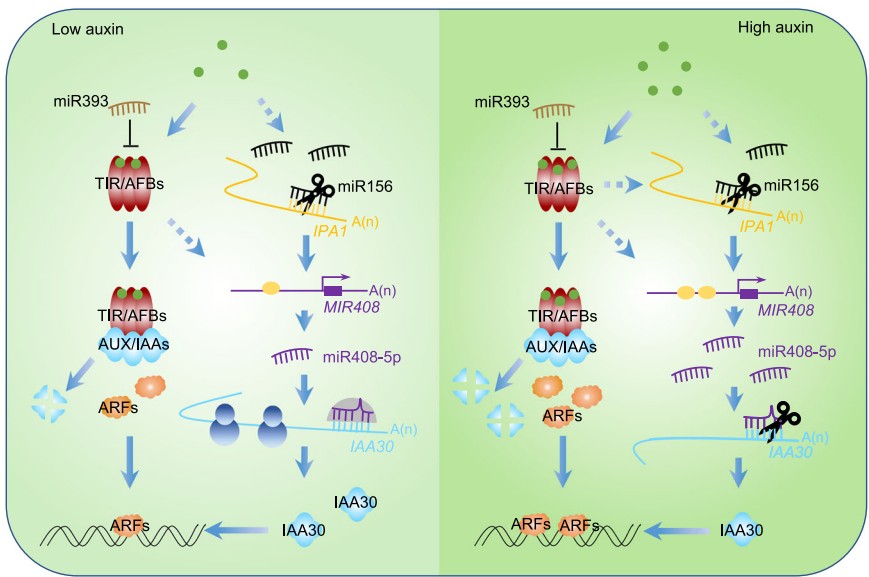

**Fig. 7 | Working model of auxin signaling involving miR408-5p regulation in rice.** Compared with that in low auxin environment, rice in high auxin condition represses miR156 and promotes more association of IPA1 with *MIR408* cis-element to trigger higher accumulations of mature miR408-5p, thereby leading to switch miR408-5p action from translation repression to mRNA cleavage. In addition to a higher contribution from canonical receptor-mediated auxin signaling, more robust regulation of *IAA30* by miR408-5p in rice through mRNA cleavage results in an enhanced auxin response in a high than in a low auxin environment.

and *like-amp1* (*lamp1*), exhibit severe pleiotropic growth defects[8,10,12]. Thus, it is difficult to conclude whether translation repression by miRNA is also a fundamental approach to target regulation as mRNA cleavage in plants. If the two silencing modes coexist and cannot be separated, the physiological importance of each mode in miRNA action remains unresolved.

In the present study, we uncover that miR408-5p in rice represses target protein translation under the ordinary circumstance, but it switches mRNA degradation, thereby enhancing regulatory efficiency when plants sense an auxin signal. Our finding separates the two mechanisms of miRNA actions temporally and demonstrates their balance is essential for rice architecture plasticity, such as leaf inclination.

To date, there have been very few reports on the functions of non-conserved miRNAs in plants, mainly due to their low abundances compared to the conserved miRNAs. Nevertheless, miR834, an *Arabidopsis*-specific miRNA, that has received more attention because it silences *CIP4* by translation repression[8]. Since miR834 is a non-conserved miRNA and almost perfectly matches with the target, it has

been proposed that young miRNA may be predominately utilize the translational repression pathway regardless of the mismatches in small RNA-target hybrids[8]. This prospective appears reasonable, because miR408-5p, which regulates *IAA30* in rice, performs translational control of IAA30 under normal environments. Nonetheless, miR408-5p can switch action modes to mRNA cleavage for *IAA30* regulation when subject to a high auxin signal. Considering that miR408-5p is present and targets *IAA30* orthologs in a few monocots, we propose that miRNAs with intermediate conservation in plants generally mediate the inhibition of target protein translation, while they may promptly change their mode of action to mRNA clearance, thereby enhancing target regulation efficiency under specific conditions.

It is noteworthy that miR408-3p is one of the most ancient miRNAs in land plants[48]. A couple of studies have shown that miR408-3p is involved in multiple physiological processes across diverse plants, including photosynthesis, fertility, biomass and tolerance against adverse environments through efficient suppression of conserved target expressions[25,26,49–52]. In the present study, we discover that miR408-5p, another mature miRNA generated from the same

precursor in rice, mediates auxin signaling cascade and development. Therefore, considering the versatile roles of *MIR408* in plants, it has a huge potential to serve as an integrator for crop improvement with multiplex desirable traits.

## Methods

### Plant material and growth conditions

Japonica rice (*Oryza sativa*) variety *Nipponbare* (NIP) was used for auxin treatment, as well as for cloning of *MIR408* and *IAA30* genes, plant transformation and protoplast isolation, unless otherwise specified. Rice seeds were germinated in sterilized water and planted in a growth chamber under a 14 h light (28 °C)/10 h dark (22 °C) cycle for the growth of transgenic plants. For auxin sensitivity assay in roots, rice seeds were surface-sterilized, and imbibed for 48 h in distilled water under dark condition at 30 °C. The synchronously germinated seeds were then grown on a fluid nutrient medium for 3 days and then supplemented with the indicated concentration of IAA for 11 days. The lengths of the primary and adventurous seedling roots of indicated lines were then measured and analyzed. For rice transformation, mature rice embryos were used and placed on N6 callus initiation medium supplemented 2.5 mg/L 2,4-D to induce callus. Rice plants for leaf angle determination and paraffin sections were grown in the paddy field with routine management in Changxing and Hangzhou in Zhejiang Province of China under natural summer field conditions. Three independent transgenic lines with a single copy of the insertions were used to observe the phenotypes unless otherwise specified. All transgenic plants analyzed in our study were $T_3$ generation homozygous plants.

### Plasmid construction

The coding sequence of *IAA30* was amplified from rice cDNA through PCR, and a 0.6 kb sequence containing *MIR408* precursor from rice DNA was cloned into a modified pCAMBIA1390 with a maize *UBI* promoter for over-expression in rice. To create the *STTM-3p* and *STTM-5p* constructs, the sequences with *KpnI* and *BamHI* sites of STTM structure against miR408-3p and miR408-5p were produced by overlapping PCR, respectively. These sequences were then cloned into the modified pCAMBIA1390. The *mir408*, *iaa30* and *iaa11/30* mutants in rice were generated by CRISPR-cas9 strategy. The sgRNA and constructs were designed and made essentially as described before[53]. For transient expression of *IAA30* in rice protoplasts, *IAA30* with its 3′UTR was amplified from NIP rice cDNA. Target site mutated IAA30 at 3′UTR (mUTR) was generated through overlapping PCR. Both Flag-tagged IAA30-UTR and Flag-tagged IAA30-mUTR were constructed with a *CaMV35S* promoter for transient overexpression in rice protoplasts.

To construct *35S::MIR408-NOS*, *IAA30pro::LUC-IAA30-3′UTR*, *IAA30pro::LUC-IAA30 m3′UTRs*, *IAA19pro::LUC-IAA19-3′UTR* and *IAA19-pro::LUC-IAA19 m3′UTR* for infection of *Nicotiana benthamiana*, *MIR408*, *IAA30 3′UTR* and *IAA19 3′UTR* were amplified from cDNA, while the promoter of *IAA30* and *IAA19* were amplified from DNA of rice plants. *MIR408* was introduced into a modified pCAMBIA2300 under the control of *CaMV35S* promoter, while *LUC-IAA30-3′UTR*, *LUC-IAA30-m3′UTRs*, *LUC-IAA19-3′UTR* and *LUC-IAA19 m3′UTR* were inserted behind their respective 2 kb native promoter. For luciferase imaging assays, the coding region of *IPA1* was obtained from NIP cDNA and inserted into pCAMBIA2300 under the control of *CaMV35S* promoter, while the promoter of *MIR408* was constructed into the modified pGreen0800 II-LUC vector, which carries the *Renilla* (*REN*) gene for calculating the relative LUC activity. *IAA30pro::LUC-IAA30-3′UTRs* and *IAA30pro::LUC-IAA30 m3′UTRs* constructs were also transformed into rice to generate *IAA30* and *IAA30m* plants, respectively.

For the *IAA30pro::GUS* and *MIR408pro::GUS* construct, the *CaMV35S* promoter in pCAMBIA1301 was replaced by a 2.1 kb promoter before the start codon of *IAA30* and a 2.9 kb promoter before the precursor of *MIR408*, respectively.

All binary expression vectors were introduced into *Agrobacterium tumefaciens* strain AGL1 for rice transformation and EHA105 for the infiltration of tobacco leaves. All primers used for plasmid construction are listed in Supplementary Data 3.

### Transfection of rice protoplasts and western blots

Generally, the plasmids *35S-Flag-IAA30-UTR* or *35S-Flag-IAA30-mUTR* were transformed into rice protoplast according to the protocols described before[54]. Briefly, Ten-day-old WT and indicated transgenic rice seedlings were used to isolate protoplasts. Around 200 μl protoplast suspension (containing ~$2 \times 10^5$ protoplasts) was transfected with 10 μg of plasmid in a 110 μl PEG solution. The transformation mixture was incubated in dark for 15 min at 28 °C, then diluted with 1 ml W5 solution (NaCl, 154 mM; CaCl$_2$, 125 mM; KCl, 5 mM; MES, 2 mM, pH 5.7) and centrifuged at 120 g for 3 min. Protoplasts were suspended in W1 solution (Mannitol, 0.5 M; KCl, 20 mM; MES, 4 mM, pH 5.7) and transferred into multi-well plates and incubated at 28 °C for 16 h. Total proteins were extracted from half of the transfected protoplasts, and the remaining protoplasts were used for RNA extraction and RT-PCR.

For immunoblot analyses, total proteins were isolated from the corresponding protoplasts of WT and transgenic seedlings. The proteins were separated by sodium dodecyl sulfate-polyacrylamide gel electrophoresis, transferred to polyvinylidene difluoride membranes, immunoblotted with corresponding commercial antibodies, and detected using High-sig ECL Western Blotting Substrate (Tanon).

### Homology based identification of miR408-5p and IAA30 target sequences in diverse plants using BLAST

Mature miRNAs and miRNA precursors for *MIR408* in 14 selected plant species were retrieved from the miRBase database (version 22). The genomic sequences were downloaded from the Ensembl Plants database (version 50) except for *citrus sinensis*, which was downloaded from the Orange (*citrus sinensis*) genome annotation project (version 2.0) (http://citrus.hzau.edu.cn/orange /download/index.php)[55]. The orthologous genes of *IAA30* among species were identified based on reciprocal best blast hits (RBH) with BlastN (Query identity = 50% and $e = 1 \times 10^{-10}$). Sequences of *MIR408* precursors and *IAA30* in the 14 selected plant species were aligned using MAFFT (version 7). The sequence alignment files were used to calculate the sequence identity at each position. An unrooted phylogenetic tree of *MIR408* precursors was constructed using RAXML with the maximum-likelihood (ML) method. The expression matrix for the two mature miRNAs from *MIR408*, *MIR166a*, *MIR167a*, *MIR168a*, *MIR319a* and *MIR396e* were retrieved from the PmiRExAt database (http://pmirexat.nabi.res.in/datashow.html)[30].

### GUS staining

Seedlings and plants of *MIR408p-GUS* and *IAA30p-GUS* transgenic rice were stained in a GUS staining solution (2 mM 5-bromo-4-chloro-3-indolyl-b-D-glucuronic acid, 50 mM NaH$_2$PO$_4$/Na$_2$HPO$_4$ buffer, 2 mM each K$_3$Fe(CN)$_6$/K$_4$Fe(CN)$_6$, 10 mM Na$_2$EDTA and 0.1% Triton X-100) and incubated at 37 °C for 8 to 12 h. After GUS staining, chlorophyll was removed using 70% ethanol.

### RT-PCR and real time qRT-PCR

Total RNAs were isolated from rice roots using the Trizol reagent (Takara) unless otherwise specified. After determining the RNA quantification, the RNAs were reverse transcribed by M-MLV (Promega) by using oligo (dT). The cDNAs were then used as templates for direct PCR or quantitative PCR. Quantitative PCR was performed using SYBR Premix (Invitrogen) on StepOnePlus™ System (Thermo-Fisher) and the *Actin* gene was used as an endogenous control. The abundance of mature miRNA was determined by the stem-loop qRT-PCR as described before and *U6* was used as an endogenous control. Three biological repeats of each qRT-PCR were conducted and each value

indicates the average with the standard error. All primer sequences are listed in Supplementary Data 3.

## Degradome sequencing and analysis

Degradome libraries were prepared from rice with high miR408-5p accumulations using three different materials. The first material that we used to isolate total RNA was seedlings of *MIR408-OE* transgenic plants, the second material was explant using primary roots under 3d 10 μM 2,4-D treatment, and the third material was 14d callus induced from rice mature embryo on $N_6$ callus initiation medium with 2.5 mg/L 2,4-D application. Degradome library construction was performed with a slight modification as previously described[56,57]. Briefly, poly (A) enriched RNA was mixed and annealed using biotinylated random primers. RNAs with 5′-monophosphates were ligated to 5′adapters and the libraries were subject to single-end sequenced using the 5′adapter on an Illumina HiSeq 2500 (LC-Bio, China). The sequencing data were analyzed using CleaveLand3.0. The degradome reads were matched to generate the degradome density file using script "CleaveLand3_map2dd.pl". Reads that mapped to the predicted *IAA30* target sites were utilized to determine the positions of the 5′transcript ends using an in-house Perl script. A couple of sets publicly available of degradome sequencing data from wild-type seedlings (PRJNA118841, PRJNA123507, PRJNA120547, PRJNA263869, PRJNA170713, PRJNA263450, PRJNA596446, PRJNA627552, PRJNA594662, PRJNA630172, PRJNA561367, PRJNA486509, PRJNA480216, PRJNA388531, PRJNA182050, PRJNA277443 and PRJNA263869) were used as control samples of rice plants without auxin treatments[58–74].

## Mapping of the *IAA30* mRNA cleavage sites by 5′RACE

For mapping the cleavage site in *IAA30* mRNA, total RNA was isolated from *MIR408-OE* transgenic plants and poly(A) + mRNA was purified by Capture mRNA Kit (Qiagen, Germany). 5′ RACE was performed according to the instruction manual provided in the Kit of 5′ RACE System for Rapid Amplification of cDNA Ends (Invitrogen life science technologies, USA). PCR products were gel purified and cloned into pMD18-T vector (Takara, Japan) for sequencing. The oligo primers are listed in Supplementary Data 3.

## Transcriptome sequencing and analysis

Synchronously germinated seeds of *STTM-5p*, *STTM-3p*, *IAA30-OE* and WT rice were grown on a fluid nutrient medium for 3 days and treated with 10 μM IAA for RNA isolation. Transcriptome libraries were constructed for each root sample individually, and then sequenced using Hiseq 2500 (LC-Bio, Hangzhou, China). Three biological replicates were conducted for each treatment.

The clean reads from each sample were aligned to rice reference genome (IRGSP-1.0) using HISAT2 software[75]. Only uniquely mapped reads were used for expression quantification with StringTie v2.0. Differential expression analysis for individual studies was performed using Ballgown package[76]. Gene expression was quantified as fragments per kb per million reads (FPKM). Genes with a fold-change greater than 2.0 and an adjusted p-value less than 0.05 were considered as differentially expressed genes (DEGs), which were subjected to GO functional enrichment analyses using AgriGO v2.0[77].

## ChIP-qPCR

The wild-type rice and *IPA1pro:: mIPA1-GFP* transgenic plants obtained from Dr. Jiayang Li's lab were used for ChIP assays according to the EpiQuikTM Plant ChIP Kit protocol (Epigentek, USA). Briefly, 2 g samples were crosslinked with 1% (v/v) formaldehyde under 10 min vacuum and then ground to powder in liquid nitrogen. The nuclei and chromatin complexes were isolated, sonicated and then incubated with anti-GFP antibody for immunoprecipitation. The precipitated DNA was recovered and dissolved in water for qPCR with the indicated primers using SYBR Premix (Invitrogen) on StepOnePlus™ System

(Thermo-Fisher). The fold enrichment was calculated between *IPA1-pro:: mIPA1-GFP* and the WT input. Fold of enrichment in *IPA1pro:: mIPA1-GFP* against the WT was calculated from three independent qPCR.

## Luciferase imaging assays

To determine the effects of miR408-5p on *IAA30* and *IAA19*, equal concentrations and volumes of *A. tumefaciens* strains EHA105 harboring indicated WT and mutated target site of *IAA30* or *IAA19* along with *MIR408* or control constructs were co-infiltrated into *N. benthamiana* leaves. At least four leaves from independent *N. benthamiana* plants were infiltrated and observed the fluorescence signal. To detect inductive effects of IPA1 on *MIR408* transcription, equal concentrations and volumes of *A. tumefaciens* strains EHA105 harboring *ProMIR408::LUC* construct with *CaMV35S::IPA1* or control constructs were infiltrated into *N. benthamiana* leaves. For observation of the fluorescence signal, the injected tobacco leaves were kept in a dark condition for 5 min after a spray of Luciferin (100 μM). The LUC images were taken by the imaging device (Tanon 5200) and the LUC signals were measured by GloMax Luminometer System (Promega, USA) and calculated by LUC/REN ratio.

## Yeast one-hybrid assays

The coding sequence of *IPA1* was amplified and subcloned into pGADT7 (Clontech). *IAA30* promoter regions P1 (−1287 to −1264 bp before precursor), P2 (−558 to −414 bp before precursor), P3 (−425 to −108 bp before precursor), P4 (−151 to 4 bp before precursor) were amplified and subcloned into pAbAi vector. The resulting constructs were then transformed into yeast strain Y1HGold. The yeast transformants were cultured on dropout (-Leu) medium containing 150−300 ng/ml Aureobasidin A (AbA) for 3 days before observation. The assay was performed with three replicates.

## Leaf angle measurement and cytological analysis

Leaf lamina joints with 1 cm sections of the flag leaf and sheath of the main tiller were cut 40 days after heading and photographed. Leaf angle was calculated by 180° minus the angle between sheath and leaf using a semicircular instrument for at least 12 leaf angles of individual plants. For cytological section analysis, the leaf lamina joints were fixed in FAA solution (40% ethanol, 5% acetic acid, and 12.5% formaldehyde in water) for 24 h and then dehydrated in a graded ethanol series and xylene-ethanol solution. Samples were embedded in paraffin for 1 day, then sections were deparaffinized in xylene, hydrated via a graded ethanol series, and stained by Fast GreenFCF after a cut into small pieces. The extra stain was rinsed, and sections were re-dehydrated by a graded ethanol series and microscopically observed. Cell numbers, length and width were calculated by ImageJ software.

## Immunoblot quantification analysis

Quantification of immunoblots was conducted according to the band intensities of Flag-IAA30 and Actin as a loading control for total lysates, which were measured with Image J. Relative band intensities were then calculated using the ratio of IAA30/Actin for each immunoblot. All immunoblot experiments were performed at least three times, essentially with the same conclusions.

## Reporting summary

Further information on research design is available in the Nature Portfolio Reporting Summary linked to this article.

# Data availability

Degradome Sequencing and RNA-seq data were deposited at Genome Warehouse in BIG Data Center under BioProjectID PRJCA005540. Source data are provided with this paper.

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

## Acknowledgements

We sincerely thank Drs. Jiayang Li, Zuhua He and Donglei Yang for sharing miR156 and IPA1 related rice seeds. We are grateful to Drs. Jiayang Li and Hong Yu for sharing with us their IPA1 ChIP-seq data. We thank Yuzhen Hu at Agriculture Experiment station of Zhejiang University for looking after rice plants. This work was supported by Zhejiang Provincial Natural Science Foundation of China (LZ21C130002), the National Natural Science Foundation of China (91940301, 31871588, 91640109, 32370341), Sanya Science and Technology Innovation Program (2022KJCX48) and the open funds of the State Key Laboratory of Rice Biology (20200404).

## Author contributions

L.W. supervised the research and designed the experiments. F.R., Xia.W., Y.Z., E.Y., S.M. and F.N. produced rice transgenic and mutant lines and F.R. carried out phenotype observation and most of other experiments. Y.L. performed miR408-5p-mediated protein translation repression experiments. P.D. and Y.S. performed bioinformatics analysis. H.B. provided miR393 overexpression rice and degradome library from explants of rice major roots. Xiangjin.W. and P.H. provided fields for rice planting. F.R., Y.L., P.D., W.Z. and L.W. analyzed the data. F.R. and P.D prepared for the figures and L.W. wrote the article.

## Competing interests

The authors declare no competing interests.
