## [Peer Review File · Nature Communications]

Reviewers' comments:

Reviewer #1 (Remarks to the Author):

In this study, authors presented quite extensive data attempted at delineating an auxin responsive pathway centred on the non-canonical miRNA, miR408-5p, in rice. In my opinion, the two most significant findings in this work were: 1) when the miR408 precursor is over expressed in rice, the IAA30 transcript was cleaved at a site complementary to miR408-5p by degradome sequencing (fig 4B) and the protein level was decreased (fig 4G), and 2) the miR408 locus was transcriptionally regulated by IPA1.

Although the topic was interesting and data presented were extensive, I found the two central arguments in this work, namely miR408-5p is a functional miRNA targeting IAA30, and that miR408-5p instead of the established miR408-3p is involved in auxin response, were flawed.

First, regarding whether miR408-5p is a functional miRNA targeting IAA30. The experimental design for fig 2B was fundamentally flawed. This type of experiments need internal controls to normalize the activity of the test promoter (e.g., using a different promoter to drive REN and calculate the LUC/REN ratio) because of the inevitable variations in the transient expression systems. The control sequence (mutated target site for miR408-5p) was usually. Typically, only a few nucleotides in the critical positions were mutated. The control sequence used by the authors was almost a completely irrelevant sequence. Rationale for using such a control sequence should be justified.

Authors claimed that 5' RACE did not detect miR408-5p directed cleavage of IAA30 transcript. Given the degradome sequencing data in fig 4B-E, I recommend authors to perform 5' RACE using these transgenic plants to confirm the sequencing results and/or to rule out technical failure in the 5' RACE assay.

Authors claimed that miR408-5p may repress IAA30 using different modes based on convenience of the individual assays. However, a consistent trend in the results observed by this reviewer was that miR408-5p did not influence IAA30 expression under most circumstances and only when the miR408 locus was over expressed, were the IAA30 transcript cleaved (fig 4B) and the protein level consistently decreased (fig 4G). Based on my reading of the results, it was difficult not to draw the conclusion that impact of miR408-5p on IAA30 was an artifact caused by over activation of the miR408 locus. Authors need to clarify this possibility.

Second, regarding a potential role of miR408-5p in auxin response, the elephant in the room was apparently miR408-3p. Unfortunately, in all genetic data presented (except where STTM was used, but this technique suffers its own drawbacks such as specificity, efficiency, and particularly in this case possible interference with the star sequences), the entire locus was used and thus any phenotype observed could be a direct or indirect effect of miR408-3p. For example, the leaf angle phenotype was reported previously in rice (e.g., supplemental fig 10 in Pan et al., 2018). In these cases, the effect was attributed to miR408-3p due to increased growth/yield. The potential discrepancy needs to be clarified with more tailed genetic analysis, such as modifying key miR408-5p nucleotides without altering miR408-3p processing. In the methods section, I did not find any description on how the transgenic lines were selected (e.g., how many independent lines were analysed, copies of the insertions, at what

generations were the phenotype assessed, etc). Rice transformation involves auxin treatment and this is important when considering auxin responses in the transgenic plants.

Reviewer #2 (Remarks to the Author):

In the manuscript entitled 'Switching action modes of miR408-5p mediates auxin signaling in rice' Rong et al. tried to understand the role of mir408 in modulating rice development through regulating auxin signaling. They also demonstrated that miR408-5p but not miR408-3p selectively targets auxin signaling through modulating IAA30 activities. They further tried to demonstrate that

miR408-5p possibly regulates both the transcriptional and translational expression of IAA30 depending on the IAA abundance. Lastly, they claimed that the mir408 regulation of auxin signaling is linked to the silencing of the TIR1 mediated by another MIR, mir393. In general, this is an interesting story. The authors performed an extensive amount of work. However, in some cases, the data interpretation was far-fetched and not directly supported by the experiments.

Major Concerns:

1. The manuscript needs extensive language editing. There are many sentences in the manuscript which are ambiguous and do not make sense.
2. The authors use 10 uM IAA for all the experiments, but did not explain why they chose the concentration. For the root growth developmental experiment, 10uM is a very high concentration, which does not match with the intracellular level of auxin. The authors should clarify this issue.
3. Figure 2C, MIR408 expression was induced 4-fold by IAA treatment, while miR 408-5p expression was induced only 2-fold by IAA application. Since MIR 408 is composed of both 5p and 3p, I assume that the rest of the expression is coming from 3-p. However, the authors claimed that mature miR408-5p is subject to transcriptional and processing control in rice. To claim this similar experiments should be performed with miR408-3p. I understand that the authors targeted the 5p as a target from computational analysis, but to confirm the biological significance of these two forms contrasting results must be shown with a similar set of experiments.
4. Figure 4, The transcriptional down-regulation and translational down-regulation do not match. Please explain the difference between I and II. I could not follow the explanation. If they are independent lines then the results are not consistent among the lines.

In anti-flag panel of Figure H, Wt signal was quantified as 2.14 but the signal does not look much different than others. How many biological replicates were used for quantification? A quantitative assay with statistical analysis is required to claim the differences. This is requested for all the protein blots.

5. ipa1 mutant and MIR156OX lines show increased primary root length but they show a reduced shoot phenotype. How these differences can be explained?

6. To convincingly demonstrate that mir408-5p acts through TIR1, instead of using MIR mutants, it is advisable to use the tir1 mutant. That will provide direct evidence that miR156-IPA1-miR408-5p loop is TIR1 dependent. The OX lines can produce many artifacts.

Reviewer #3 (Remarks to the Author):

In the manuscript, Rong et al. reported that miR408-5p could target IAA30, a critical gene in auxin signaling, and it had different action modes under conditions of different level of auxin levels. It also found that the miR156-IPA1 module could regulate the MIR408 to mediate leaf inclination. Overall, the manuscript was well written and results were represented. However, there are some weaknesses that need to be addressed.

(1) Since the single MIR408 precursor can produce both miR408-5p and miR408-3p, which targets IAA30 and UCL genes, respectively. Since both miRNAs are functional, I was keeping thinking about how to separate the function of these two miRNAs completely. In other words, how to mute the function of one without affecting the other one. Although the STTM technique was used for this purpose, I am wondering that it may be still possible that the silencing of the miRNA strand using STTM (in this study, STTM-5p) could affect the processing of MIRNA precursor and then the production of the miRNA* strand (STTM-3p). So I would recommend to check the abundance of MIR408-3p in the STTM-5p lines to confirm that it is not changed.

(2) I feel it is superfluous to include the miR393-TIR module in the regulatory circuit as there is no direct connection between miR393-TIR and the newly identified module, except the auxin signaling network. But as the auxin is such an important signal in plants and TIR is the receptor, so the miR393-TIR module may be associated with all kinds of regulatory circuits. There are other miRNA-involved pathways related to auxin signaling, like miR390-TAS3-ARF, miR160-ARF and miR167-ARF. I would argue that the miR408-5p-IAA30 may respond to the silencing or break-down of any of these pathways.

(3) In lines 118-120, regarding the “evolution”, these are only alignment to show sequence variation, not real “evolution”. I don’t know what it means by “miR408-3p probably retained earlier than miR408-5p”, I believe the miR/miR* duplex are co-evolved. How can they be separately retained.

Responses to Reviewers' Comments

We are grateful for the editor and reviewers' interest in our research and we would like to express our sincere appreciation for their constructive comments to help us improve our manuscript. We have performed additional experiments, added discussions, improved language and accordingly revised the manuscript. We hope that we have adequately addressed all the concerns raised by the reviewers and that the manuscript is ready for publication. The point-by-point responses to the reviewers' comments are listed in detail below:

Reviewer #1

(From the reviewer) In this study, authors presented quite extensive data attempted at delineating an auxin responsive pathway centered on the non-canonical miRNA, miR408-5p, in rice. In my opinion, the two most significant findings in this work were: 1) when the miR408 precursor is over expressed in rice, the IAA30 transcript was cleaved at a site complementary to miR408-5p by degradome sequencing (fig 4B) and the protein level was decreased (fig 4G), and 2) the miR408 locus was transcriptionally regulated by IPA1.

Although the topic was interesting and data presented were extensive, I found the two central arguments in this work, namely miR408-5p is a functional miRNA targeting IAA30, and that miR408-5p instead of the established miR408-3p is involved in auxin response, were flawed.

First, regarding whether miR408-5p is a functional miRNA targeting IAA30. The experimental design for fig 2B was fundamentally flawed. This type of experiments need internal controls to normalize the activity of the test promoter (e.g., using a different promoter to drive REN and calculate the LUC/REN ratio) because of the inevitable variations in the transient expression systems. The control sequence (mutated target site for miR408-5p) was usually. Typically, only a few nucleotides in the critical positions were mutated. The control sequence used by the authors was almost a completely irrelevant sequence. Rationale for using such a control sequence should be justified.

Response: Thank you for the reviewer's interest in our work. We acknowledge and agree with the reviewer's comment regarding the experimental design for Fig 2B. It is important to include a control to normalize the activity of the test promoter. As suggested by the reviewer, we have addressed the concerns by creating new constructs that include REN as a control to calculate the ratio of LUC/REN to avoid inevitable variations in the transient expression system. We have performed the experiment again. We observed a significant reduction in the relative LUC signal when miR408-5p was co-expressed with *IAA30*, while the signal remained similar in the co-expression of miR408-5p and mutated *IAA30* in *N. benthamiana* leaves (revised Figure 2B, revised Figures S3A and S3B). These results confirm our previous findings that *IAA30* is targeted by miR408-5p.

To address second concern raised by the reviewer, we introduced the mutations in the nucleotides either at 5' the arm or at the 3' arm of the *IAA30* target sites and repeated the experiments. We observed that both mutated versions of *IAA30* were able to escape regulation by miR408-5p (revised Figures S3A and S3B), suggesting that regulation of *IAA30* by miR408-5p is specific.

(From the reviewer) Authors claimed that 5' RACE did not detect miR408-5p directed cleavage of *IAA30* transcript. Given the degradome sequencing data in fig 4B-E, I recommend authors to perform 5' RACE using these transgenic plants to confirm the sequencing results and/or to rule out technical failure in the 5' RACE assay.

Response: As the reviewer suggested, we performed 5'RACE using *MIR408-OE* transgenic plants. The results of 5'RACE confirmed that the cleavage of *IAA30* by miR408-5p occurred between 8th and 9th nucleotide of the miRNA, which is consistent with the degradome data showing cleavage of *IAA30* by miR408-5p at a position 1 nucleotide upstream of the conventional cleavage site (revised Figure S14A).

(From the reviewer) Authors claimed that miR408-5p may repress *IAA30* using

different modes based on convenience of the individual assays. However, a consistent trend in the results observed by this reviewer was that miR408-5p did not influence IAA30 expression under most circumstances and only when the miR408 locus was over expressed, were the IAA30 transcript cleaved (fig 4B) and the protein level consistently decreased (fig 4G). Based on my reading of the results, it was difficult not to draw the conclusion that impact of miR408-5p on IAA30 was an artifact caused by over activation of the miR408 locus. Authors need to clarify this possibility.

Response: Thank you for bringing this to our attention. Actually, our conclusion regarding the regulation of *IAA30* by miR408-5p was demonstrated by the enhanced IAA30 protein level in *mir408* mutants and *STTM-5p* plants, in which miR408-5p disappeared or its activity was repressed, in addition to the decrease of IAA30 expression and protein in *MIR408* overexpression lines. Moreover, we found that the protein level of IAA30 was not increased in *STTM-3p* plants, in which the activity of miR408-3p was inhibited (revised Figure 4), suggesting that IAA30 is not regulated by miR408-3p.

In addition, as reviewer 3 suggested, we examined the accumulation of miR408-3p in *STTM-5p* plants and miR408-5p in *STTM-3p* plants by stem-loop qRT-PCR, and found neither miR408-3p nor miR408-5p could be affected in *STTM-5p* plants and *STTM-3p* lines, respectively, suggesting that blocking miR408-5p and miR408-3p by *STTM-5p* and *STTM-3p* was specific (revised Figures S9A and S9B). Therefore, the regulation of miR408-5p on IAA30 is not an artifact.

(From the reviewer) Second, regarding a potential role of miR408-5p in auxin response, the elephant in the room was apparently miR408-3p. Unfortunately, in all genetic data presented (except where *STTM* was used, but this technique suffers its own drawbacks such as specificity, efficiency, and particularly in this case possible interference with the star sequences), the entire locus was used and thus any phenotype observed could be a direct or indirect effect of miR408-3p. For example, the leaf angle phenotype was reported previously in rice (e.g., supplemental fig 10 in Pan et al., 2018). In these cases, the effect was attributed to miR408-3p due to

increased growth/yield. The potential discrepancy needs to be clarified with more tailed genetic analysis, such as modifying key miR408-5p nucleotides without altering miR408-3p processing. In the methods section, I did not find any description on how the transgenic lines were selected (e.g., how many independent lines were analysed, copies of the insertions, at what generations were the phenotype assessed, etc). Rice transformation involves auxin treatment and this is important when considering auxin responses in the transgenic plants.

Response: Yes, the previous study by Pan et al., reported a decrease in leaf angle in *MIR408* precursor overexpression plants, but they did not figure out which mature miRNA derived from *MIR408* was responsible for this process ¹. On the other hand, while we did not observe a significant reduction in leaf angle in our *MIR408-OE* plants, possibly due to differences in plant growth conditions between the two studies, we did find a notable increase in leaf inclination in *mir408* plants and *STTM-5p* plants but not in *STTM-3p* plants (revised Figure 6), indicating that miR408-5p derived from *MIR408* plays a crucial role in controlling leaf angle.

We appreciate the suggestion from the reviewer regarding the genetic analysis of modifying key miR408-5p nucleotides without altering miR408-3p processing is useful to clarify the functions of miR408-5p. However, we would like to say that the technique for this is impossible thus far. On the other hand, we utilized the STTM approach to specifically block the activity of miR408-5p and miR408-3p (stated above, Figures S9A and S9B), and observed that only *STTM-5p* exhibited significant phenotypes in response to auxin treatment and leaf angle development (revised Figure 3 and Figure 6). These results suggest that miR408-5p, rather than miR408-3p, is involved in the regulation of auxin signaling through its targeting of *IAA30*.

Sorry, we might have missed some information regarding the transgenic plants in the methods section. We analyzed the phenotype from at least three independent transgenic lines, each with a single copy of the insertions. All of these transgenic plants analyzed in our study are T₃ generation homozygous plants. We have clearly described this in the revised materials and methods section. It is important to note that the phenotypes observations we made in response to auxin treatment were

conducted using T₄ generation transgenic plants, rather than T₀ generation transgenic plants. Therefore, these observations do not pertain to the rice transformation process.

Reviewer #2

(From the reviewer) In the manuscript entitled `Switching action modes of miR408-5p mediates auxin signaling in rice` Rong et al. tried to understand the role of mir408 in modulating rice development through regulating auxin signaling. They also demonstrated that miR408-5p but not miR408-3p selectively targets auxin signaling through modulating IAA30 activities. They further tried to demonstrate that miR408-5p possibly regulates both the transcriptional and translational expression of IAA30 depending on the IAA abundance. Lastly, they claimed that the mir408 regulation of auxin signaling is linked to the silencing of the TIR1 mediated by another MIR, mir393. In general, this is an interesting story. The authors performed an extensive amount of work. However, in some cases, the data interpretation was far-fetched and not directly supported by the experiments.

Major Concerns:

1. The manuscript needs extensive language editing. There are many sentences in the manuscript which are ambiguous and do not make sense.

Response: We improved the language in the revised manuscript via an English editing service. Thank you for pointing this out.

(From the reviewer) 2. The authors use 10 uM IAA for all the experiments, but did not explain why they chose the concentration. For the root growth developmental experiment, 10uM is a very high concentration, which does not match with the intracellular level of auxin. The authors should clarify this issue.

Response: We used a concentration of 10 uM IAA based on several previous references^{2, 3, 4}. Additionally, in response to the reviewer's comment regarding the high concentration of auxin used in the study, we also conducted experiments using lower concentration of 1 uM auxin. We observed the phenotype of *mir408*, *STTM-5p*,

and *STTM-3p* plants in response to 1 μ M auxin treatment (revised Figures S9 and S10) and found that *mir408* and *STTM-5p* (revised Figures S9D-S9G), rather than *STTM-3p* plants (revised Figures S10E and S10F), were insensitive to 1 μ M auxin treatment, which is consistent with the previous results from 10 μ M IAA treatment.

(From the reviewer) 3. Figure 2C, MIR408 expression was induced 4-fold by IAA treatment, while miR 408-5p expression was induced only 2-fold by IAA application. Since MIR408 is composed of both 5p and 3p, I assume that the rest of the expression is coming from 3-p. However, the authors claimed that mature miR408-5p is subject to transcriptional and processing control in rice. To claim this similar experiments should be performed with miR408-3p. I understand that the authors targeted the 5p as a target from computational analysis, but to confirm the biological significance of these two forms contrasting results must be shown with a similar set of experiments.

Response: We agree with the reviewer's comment. As suggested, we examined the accumulation of miR408-3p in response to auxin treatment and conducted additional experiments. We found that both miR408-5p and miR408-3p were affected by auxin treatment, although the fold increase was different for each (revised Figures 2E and 2F). In addition, we also examined the expression of miR408-3p in *MIR393* overexpression lines and *tir1* mutants. We found that miR408-3p, similar to miR408-5p, was regulated by the miR393-TIR1/AFBs module (revised Figures 17C and 17D). Together, we consider that mature miR408-5p can be regulated at transcriptional and processing control levels in rice.

(From the reviewer) 4. Figure 4, The transcriptional down-regulation and translational down-regulation do not match.

Please explain the difference between I and II. I could not follow the explanation. If they are independent lines then the results are not consistent among the lines.

In anti-flag panel of Figure H, Wt signal was quantified as 2.14 but the signal does not look much different than others. How many biological replicates were used for

quantification? A quantitative assay with statistical analysis is required to claim the differences. This is requested for all the protein blots.

Response: Sorry for the confusion. In Fig 4, the difference between I and II indicates that the original or mutated target site in the 3' UTR of the *IAA30* gene we introduced in the experiment.

We checked the abundance of the IAA protein in anti-flag panel of Fig 4H, and found that the WB signal was indeed quantified as 2.14. We calculated the relative abundance of the protein based on the analysis results of transfection efficiency using the *hptII* gene expression and the endogenous control protein actin. We performed similar experiments with three replicates, and we included the statistical analysis in the revised Figures S14B and S14C.

(From the reviewer) 5. *ipa1* mutant and *MIR156OX* lines show increased primary root length but they show a reduced shoot phenotype. How these differences can be explained?

Response: The development of plant roots and shoots is regulated by numerous factors and involves diverse mechanisms. Therefore, shoot development is not always directly correlated with root development, even though the increased root growth can facilitate nutrients absorption and support shoot development. *IPA1* and *MIR156* likely influence shoot and root development through the regulation of different transcription factors or other proteins. This can result in an increase in primary root length but a reduction in shoot growth, as observed in the *ipa1* mutant and *MIR156OX* plants. The underlying mechanism of this process is intriguing and warrants further investigation in the future.

(From the reviewer) 6. To convincingly demonstrate that mir408-5p acts through TIR1, instead of using MIR mutants, it is advisable to use the *tir1* mutant. That will provide direct evidence that miR156-IPA1-miR408-5p loop is TIR1 dependent. The OX lines can produce many artifacts.

Response: Our collaborator, Dr. Hongwu Bian, provided us with *tir1* mutants ⁵, which

we used to examine the expression of *MIR408*, miR408-5p and miR408-3p. In most cases, the expressions of miR408-5p module component were similar in *MIR393-OE* plants and *tir1* mutants (revised Figure S17), suggesting that the miR408-5p module is regulated by the canonical TIR1/AFBs auxin receptors.

Reviewer #3

(From the reviewer) In the manuscript, Rong et al. reported that miR408-5p could target IAA30, a critical gene in auxin signaling, and it had different action modes under conditions of different level of auxin levels. It also found that the miR156-IPA1 module could regulate the *MIR408* to mediate leaf inclination. Overall, the manuscript was well written and results were represented. However, there are some weaknesses that need to be addressed.

(1) Since the single *MIR408* precursor can produce both miR408-5p and miR408-3p, which targets IAA30 and UCL genes, respectively. Since both miRNAs are functional, I was keeping thinking about how to separate the function of these two miRNAs completely. In other words, how to mute the function of one without affecting the other one. Although the STTM technique was used for this purpose, I am wondering that it may be still possible that the silencing of the miRNA strand using STTM (in this study, STTM-5p) could affect the processing of *MIRNA* precursor and then the production of the miRNA* strand (STTM-3p). So I would recommend to check the abundance of *MIR408-3p* in the STTM-5p lines to confirm that it is not changed.

Response: That is indeed a valuable suggestion to examine the accumulations of miR408-3p and miR408-5p in *STTM-5p* and *STTM-3p* plants to validate the specificity of the STTM approach. As mentioned in our response to reviewer 1 and as shown in the revised Figures S9A and S9B, we found that neither miR408-3p nor miR408-5p was affected in *STTM-5p* plants and *STTM-3p* plants, respectively, suggesting that the blocking of miR408-5p and miR408-3p in *STTM-5p* and *STTM-3p* lines was specific. We appreciate the reviewer's comment.

(From the reviewer) (2) I feel it is superfluous to include the miR393-TIR module in

the regulatory circuit as there is no direct connection between miR393-TIR and the newly identified module, except the auxin signaling network. But as the auxin is such an important signal in plants and TIR is the receptor, so the miR393-TIR module may be associated with all kinds of regulatory circuits. There are other miRNA-involved pathways related to auxin signaling, like miR390-TAS3-ARF, miR160-ARF and miR167-ARF. I would argue that the miR408-5p-IAA30 may respond to the silencing or break-down of any of these pathways.

Response: Yes, we agree with the reviewer that miR408-5p-IAA30 may affect many modules in auxin signaling, including miR390-TAS3-ARF, miR160-ARF, miR167-ARF and so on. Because TIR1/AFBs have been shown as one class of auxin receptors and the miR393-TIR1/AFBs module has been revealed to function in the regulation of auxin perception, we determined its effects on the miR408-5p-IAA30 module (Figure 7). As expected, we found that the miR393-TIR1/AFBs module indeed had roles in the modulation of miR408-5p-IAA30 in auxin signaling. Nevertheless, we cannot exclude the possibility that miR390-TAS3-ARF, miR160-ARF, and miR167-ARF modules could affect or be affected by miR408-5p-IAA30. Future studies are required to evaluate their possible regulatory loops.

(From the reviewer) (3) In lines 118-120, regarding the “evolution”, these are only alignment to show sequence variation, not real “evolution”. I don’t know what it means by “miR408-3p probably retained earlier than miR408-5p”, I believe the miR/miR* duplex are co-evolved. How can they be separately retained.

Response: We apologize for the confusion. To provide a clearer statement, we have revised it as follow: “The higher number of identical sequences in miR408-3p compared to miR408-5p indicates that miR408-3p is more conserved than miR408-5p.”

References:

1. Pan JW, *et al.* Overexpression of microRNA408 enhances photosynthesis, growth, and seed yield in diverse plants. *Journal of Integrative Plant Biology* **60**, 323-340 (2018).

2. Yamauchi T, *et al.* Fine control of aerenchyma and lateral root development through AUX/IAA- and ARF-dependent auxin signaling. *Proceedings of the National Academy of Sciences of the United States of America* **116**, 20770-20775 (2019).
3. Chen Y, Yang QF, Sang SH, Wei ZY, Wang P. Rice Inositol Polyphosphate Kinase (OsIPK2) Directly Interacts with OsIAA11 to Regulate Lateral Root Formation. *Plant and Cell Physiology* **58**, 1891-1900 (2017).
4. Qu L, Lin LB, Xue HW. Rice miR394 suppresses leaf inclination through targeting an F-box gene, LEAF INCLINATION 4. *Journal of Integrative Plant Biology* **61**, 406-416 (2019).
5. Guo F, *et al.* Functional analysis of auxin receptor OsTIR1/OsAFB family members in rice grain yield, tillering, plant height, root system, germination, and auxinic herbicide resistance. *New Phytologist* **229**, 2676-2692 (2021).

Reviewers' comments:

Reviewer #1 (Remarks to the Author):

I stand with my original assessment of the work by Wu and colleagues. Authors presented quite extensive data attempted at delineating an auxin responsive pathway centered on the non-canonical and putative miRNA, miR408-5p, in rice. I think the two most significant findings in this work were: 1) the rice MIR408 locus appears to produce two miRNAs, when the miR408 precursor is over expressed, the IAA30 transcript was cleaved at a site complementary to miR408-5p, and 2) the miR408 locus was transcriptionally regulated by IPA1. In the revised manuscript, authors added more experimental data to support miR408-5p targeting IAA30 in rice. This inclusion no doubt strengthened the work. However, authors still failed to address the more critical question regarding miR408-3p. The conserved miR408-3p has been shown in diverse plant species to target two types of copper proteins, the uclacyanin type and the laccase type. In rice, it has been shown that miR408-3p, via targeting uclacyanin, affects grain size and yield (e.g., Zhang et al. *Plant Physiol* 2017, 175(3):1175); Pan et al. *JIPB* 2018, 60(4):323; Yang et al. *Plant Physiol* 2021, 186(1):519). In maize and poplar, it has been shown that miR408-3p, via targeting laccases, affects the secondary cell wall (Qin et al. *Plant Physiol*. 2023, 192(2):1569; Guo et al. *Nat Commun*. 2023, 14(1):4285). Given this wealth of literature, it would be extremely odd for authors not to observe any effects of miR408-3p in their MIR408-OE plants. I previously asked authors to provide more description on how the transgenic lines were made. Was the same precursor sequence over-expressed in comparison to the previously reported transgenic lines? Have authors analysed copy number of insertions? If different constructs were used, this may affect how the miRNAs were processed and result in phenotypic discrepancies. This issue was not adequately addressed by authors.

Reviewer #2 (Remarks to the Author):

The authors tried to address my concerns in my earlier review through additional experiments. I would like to thank the authors for their efforts. The additional experimental results clearly confirmed my concerns. miR408-5p and miR408-3p were affected by auxin treatment suggesting that this response is not miR408-5p specific. The claim that miR408-5p module is regulated by the canonical TIR1/AFBs auxin receptors is also partially correct. The results also confirm that miR408-3p module is also regulated by the TIR1 pathway. These results clearly raised the question about the specificity of the miR408-5p and go against the central message of the manuscript.

Reviewer #3 (Remarks to the Author):

Thanks to the revisions and most of my concerns have been addressed. But regarding the last piece of results of the miR393-TIR module, I still think it is not a big finding, a little too much. It seems that the

authors might misunderstand my comment on this. What I meant is that, since TIR/AFBs are auxin receptor, anything affecting its activity will definitely influence the downstream auxin signaling and subsequent phenotype development. In other words, miR408-5p-IAA module is an integral part and a downstream process after auxin perception. Its response to miR393 overexpression and tir mutation is consistent with logic.

In the second-to-last paragraph, “implying that the regulation of mature miRNAs derived from MIR408 by auxin may occur at the transcriptional level”, are there any other levels of regulation? I feel the transcriptional regulation is the only possible one? “These data collectively show that auxin receptors are essential for the induction of miR408-5p by auxin”, I would say auxin receptors are essential to all auxin-related/associated pathways/processes.

Overall, I would not recommend to include the miR393-TIR part in the whole story. By the way, the regulation of miR393-TIR on miR408-IAA module is not necessarily through the IPA1 (miR156). The regulation might be via many other components. So the linear illustration of “miR393-TIR-miR156-IPA-miR408” is likely far-fetched.

Responses to Reviewers' Comments

We are grateful for all reviewers' comments that help us improve our work further. We would like to express that our major point was to present the participation of miR408-5p in auxin signaling via switching action modes to *IAA30*, but neither to ignore the role of miR408-3p in plant development, nor to exclude the possibility that miR408-3p may be also involved in auxin response in rice. To avoid misunderstanding, we weakened the statement regarding the specific significance of miR408-5p in auxin signaling in the revised manuscript. In addition, we performed more experiments and provided explanations to address the concerns raised by the reviewers. We hope that we have adequately addressed all the concerns raised by the reviewers and the revised manuscript is sufficient for publication. The point-by-point responses to the reviewers' comments are listed in detail below:

Reviewer #1 (Remarks to the Author):

I stand with my original assessment of the work by Wu and colleagues. Authors presented quite extensive data attempted at delineating an auxin responsive pathway centered on the non-canonical and putative miRNA, miR408-5p, in rice. I think the two most significant findings in this work were: 1) the rice MIR408 locus appears to produce two miRNAs, when the miR408 precursor is over expressed, the *IAA30* transcript was cleaved at a site complementary to miR408-5p, and 2) the miR408 locus was transcriptionally regulated by *IPA1*. In the revised manuscript, authors added more experimental data to support miR408-5p targeting *IAA30* in rice. This inclusion no doubt strengthened the work. However, authors still failed to address the more critical question regarding miR408-3p. The conserved miR408-3p has been shown in diverse plant species to target two types of copper proteins, the uclacyanin type and the laccase type. In rice, it has been shown that miR408-3p, via targeting uclacyanin, affects grain size and yield (e.g., Zhang et al. *Plant Physiol* 2017, 175(3):1175); Pan et al. *JIPB* 2018, 60(4):323; Yang et al. *Plant Physiol* 2021, 186(1):519). In maize and poplar, it has been shown that miR408-3p, via targeting laccases, affects the secondary cell wall (Qin et al. *Plant Physiol*. 2023, 192(2):1569; Guo et al. *Nat Commun*.

2023, 14(1):4285). Given this wealth of literature, it would be extremely odd for authors not to observe any effects of miR408-3p in their MIR408-OE plants. I previously asked authors to provide more description on how the transgenic lines were made. Was the same precursor sequence over-expressed in comparison to the previously reported transgenic lines? Have authors analysed copy number of insertions? If different constructs were used, this may affect how the miRNAs were processed and result in phenotypic discrepancies. This issue was not adequately addressed by authors.

Response: Thanks for the reviewer claiming that our additional experiments strengthened the work. For the concern of the role of miR408-3p in rice, we would like to claim that our major point was not to ignore the significant and the conserved role of miR408-3p in rice development via regulation of uclacyanin and laccase family genes. Also, we could not completely exclude the possibility that miR408-3p may be involved in auxin signaling as well through unknown mechanisms. In our work, we just attempted to present the involvement of miR408-5p in auxin signaling via switching action modes and regulation of target *IAA30*.

Similarly, because of our major point on miR408-5p, we did not describe the other traits in *MIR408-OE* plants in our previous manuscript. Actually, we also observed the bigger grain size in *MIR408-OE* plants compared with WT as reported before (Figure 1 below)^{1,2}. Meanwhile, we would like to thank the reviewer for providing us the comprehensive references regarding miR408-3p, and we referred to all of them in our revised manuscript.

Figure 1 The grain size is enhanced in *MIR408* overexpression transgenic rice

(A) The grain phenotypes of WT and *MIR408* overexpression transgenic lines.

(B) The grain length and grain width of WT and *MIR408* overexpression transgenic lines.

In the material and methods section of our last version of manuscript, we described that we made several independent transgenic lines and selected three lines with single copy insertion for further experiments based on their 3:1 segregation when they were T₁ generation. To further validate this, during the past month, we used the known single copy gene *SPS1* (*Sucrose Phosphate Synthase 1*) as a control³, and validated that all transgenic lines we used in the experiments indeed had single copy insertion through performing qRT-PCR experiments⁴ (Figure 2 below).

Figure 2 The calculated copy number of *SPS1* and *HptII* genes in WT and *MIR408* overexpression transgenic lines

In addition, as the reviewer suggested, we checked all sequences for *MIR408* overexpression transformation in the previous and our present study, and found that all of them contained the core miR408 precursor sequence^{1,2,5}. Nevertheless, since each study was performed independently several years ago, the *MIR408* sequences constructed for transformation were not fully identical (Figure 3 below), which may lead to some phenotypic discrepancies observed in different studies. Thanks the reviewer for reminding us and pointing this out.

Figure 3 A Schematic diagram showing the information of sequences used for *MIR408* overexpression in rice in the previous and present studies

Reviewer #2 (Remarks to the Author):

The authors tried to address my concerns in my earlier review through additional experiments. I would like to thank the authors for their efforts. The additional experimental results clearly confirmed my concerns. miR408-5p and miR408-3p were affected by auxin treatment suggesting that this response is not miR408-5p specific. The claim that miR408-5p module is regulated by the canonical TIR1/AFBs auxin receptors is also partially correct. The results also confirm that miR408-3p module is also regulated by the TIR1 pathway. These results clearly raised the question about the specificity of the miR408-5p and go against the central message of the manuscript.

Response: We agree the reviewer that the response to auxin was not miR408-5p specific. Because of the transcriptional regulation of their precursor, both miR408-5p and miR408-3p were changed by auxin treatment and controlled by TIR1/AFBs auxin receptors. However, this cannot support a conclusion that miR408-3p plays the major role in auxin response or is involved in auxin response

via regulation of *IAA30*.

As we present above, our major point was to present the role of miR408-5p in auxin signaling via switching action modes and regulating *IAA30*, but not to exclude the possibility that miR408-3p might be involved in auxin response. On the one hand, we did not observe a significant alternation of rice root plasticity and gene expressions in *STTM-3p* plants compared with WT when subject to auxin treatment (Supplemental Figure 9 and 10 in the manuscript), suggesting that miR408-3p may not play essential roles in auxin pathway. On the other hand, we examined the expression of *IAA30* in WT and *STTM-3p* plants and found that they were similar (Figure 4 below), suggesting that miR408-3p is not required for *IAA30* regulation. Collectively, these results indicated miR408-5p is the major player involved in auxin signaling through regulation of *IAA30*, although we could not exclude the possibility that miR408-3p may also participate in auxin pathway in rice.

Figure 4 *IAA30* is not regulated by miR408-3p in rice

(A) *IAA30* expression is comparable in WT and *STTM-3p* transgenic plants.

(B) *IAA30* responses to 10 μ M IAA treatment are similar in WT and *STTM-3p* transgenic plants.

Reviewer #3 (Remarks to the Author):

Thanks to the revisions and most of my concerns have been addressed. But regarding the last piece of results of the miR393-TIR module, I still think it is not a big finding, a little too much. It seems that the authors might misunderstand my comment on this. What I meant is that, since TIR/AFBs are auxin receptor, anything affecting its activity will definitely influence the downstream auxin signaling and subsequent phenotype development. In other words, miR408-5p-IAA module is an integral part and a downstream process after auxin perception. Its response to

miR393 overexpression and tir mutation is consistent with logic.

In the second-to-last paragraph, “implying that the regulation of mature miRNAs derived from MIR408 by auxin may occur at the transcriptional level”, are there any other levels of regulation? I feel the transcriptional regulation is the only possible one? “These data collectively show that auxin receptors are essential for the induction of miR408-5p by auxin”, I would say auxin receptors are essential to all auxin-related/associated pathways/processes.

Overall, I would not recommend to include the miR393-TIR part in the whole story. By the way, the regulation of miR393-TIR on miR408-IAA module is not necessarily through the IPA1 (miR156). The regulation might be via many other components. So the linear illustration of “miR393-TIR-miR156-IPA-miR408” is likely far-fetched.

Response: Thank you for the reviewer’s constructive comments. We agree the reviewer that “auxin receptors are essential to all auxin-related/associated pathways/processes”. Indeed, similarly, miR408-5p and its precursor are regulated by auxin receptors. We also agree that the transcriptional regulation of miR408-5p-IAA30 and miR156-IPA1 by miR393-TIR is not critical in our story because their alteration in *MIR393-OE* plants and *tir1* mutants might be indirect. Therefore, we moved these relevant figures to supplemental figures in this version of manuscript. If the reviewer persists in excluding the miR393-TIR part in the whole story, we will delete them later. In addition, as the reviewer suggested, we did not use the name of “miR393-TIR-miR156-IPA-miR408” module any more in our revised manuscript.

References:

1. Pan, J.W. *et al.* Overexpression of microRNA408 enhances photosynthesis, growth, and seed yield in diverse plants. *Journal of Integrative Plant Biology* **60**, 323-340 (2018).
2. Zhang, J.P. *et al.* MiR408 Regulates Grain Yield and Photosynthesis via a Phytocyanin Protein. *Plant Physiology* **175**, 1175-1185 (2017).
3. ValdezAlarcon, J.J., Ferrando, M., Salerno, G., JimenezMoraila, B. & HerreraEstrella, L. Characterization of a rice sucrose-phosphate synthase-encoding gene. *Gene* **170**, 217-222 (1996).
4. Yang, L.T. *et al.* Estimating the copy number of transgenes in transformed rice by real-time quantitative PCR. *Plant Cell Reports* **23**, 759-763 (2005).

5. Yang, X.F. *et al.* OsmiR396/growth regulating factor modulate rice grain size through direct regulation of embryo-specific miR408. *Plant Physiology* **186**, 519-533 (2021).